# Geometric Knowledge Distillation:
# Topology Compression for Graph Neural Networks

**Chenxiao Yang, Qitian Wu, Junchi Yan**[*]
Department of Computer Science and Engineering
MoE Key Lab of Artificial Intelligence
Shanghai Jiao Tong University
{chr26195,echo740,yanjunchi}@sjtu.edu.cn

## Abstract

We study a new paradigm of knowledge transfer that aims at encoding graph topological information into graph neural networks (GNNs) by distilling knowledge from a teacher GNN model trained on a complete graph to a student GNN model operating on a smaller or sparser graph. To this end, we revisit the connection between thermodynamics and the behavior of GNN, based on which we propose Neural Heat Kernel (NHK) to encapsulate the geometric property of the underlying manifold concerning the architecture of GNNs. A fundamental and principled solution is derived by aligning NHKs on teacher and student models, dubbed as Geometric Knowledge Distillation. We develop non-parametric and parametric instantiations and demonstrate their efficacy in various experimental settings for knowledge distillation regarding different types of privileged topological information and teacher-student schemes.

## 1 Introduction

Modern graph neural networks (GNNs) [28; 49; 53] have shown remarkable performance in learning representations for structured instances. From the perspective of geometric deep learning [5; 4; 38], part of the achievement of GNNs can be attributed to their implementation of the permutation invariance property as *geometric priors* [2] into the architecture design. Nevertheless, in practice, GNNs highly rely on graph topology, as essential input information, to explore the relational knowledge implicit in interactions of instance pairs throughout the entire message passing process, termed as *geometric knowledge* in this paper. As advances in generalized distillation [33; 47] reveal the possibility of encoding input features into model construction, natural questions arise as to:

*Is it possible, and if so, how can we encode graph topology as a special type of 'geometric prior' into a GNN model, such that the model could precisely capture the underlying geometric knowledge even without full graph topology as input?*

In specific, we are interested in the following *geometric knowledge transfer* problem: a GNN model (with node-specific outputs for node-level prediction [23]) is exposed with a partial graph, which is a subset of the complete graph. Formally speaking, we have notations:

$$\mathcal{G} = \{\mathcal{V}, \mathcal{E}\} \text{ (partial graph)}, \ \tilde{\mathcal{G}} = \{\tilde{\mathcal{V}}, \tilde{\mathcal{E}}\} \text{ (complete graph)}, \text{ where } \mathcal{V} \subseteq \tilde{\mathcal{V}}, \mathcal{E} \subseteq \{\mathcal{V} \times \mathcal{V}\} \cap \tilde{\mathcal{E}}. \quad (1)$$

Our goal is to transfer or encode geometric knowledge extracted from $\tilde{\mathcal{G}}$ to the target GNN model that is only aware of $\mathcal{G}$. Studying this problem is also of much practical value. As a non-exhaustive

---

[*]Junchi Yan is the correspondence author who is also with Shanghai AI Laboratory.

[2]Geometric priors originally refer to the geometric principles naturally encoded in deep learning architectures, e.g., translational symmetry for CNNs, permutation invariance for GNNs.

36th Conference on Neural Information Processing Systems (NeurIPS 2022).

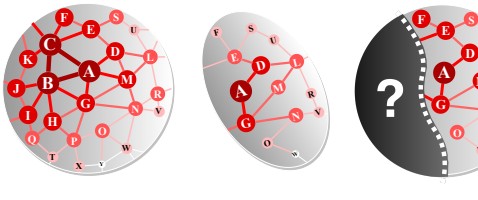

a) Teacher     b) Student (before GKD)     c) Student (after GKD)

Figure 1: Feature propagation on the underlying manifold $\mathcal{M}$. (a) Teacher: aware of the complete graph topology, and faithfully explore geometric knowledge about the underlying manifold. (b) Student before GKD: only aware of partial graph topology, and estimate biased geometry property. (c) Student after GKD: able to propagate features on the same space as teacher by alignment of NHKs.

list of applications: improving efficiency without compromising on effectiveness for coarsened graphs [14; 24; 69], privacy constrained scenarios in social recommenders or federated learning where the complete graph is unavailable [34; 50; 68], promoting concentration on targeted community to bring up economic benefits [57].

Achieving this target is non-trivial for that we need to first find a principled and fundamental way to encapsulate the geometric knowledge extracted by GNN model, which requires in-depth investigation on the role of graph topology throughout the progressive process of message passing. Therefore, we take a thermodynamic view borrowed from physics and propose a new methodology built upon recent advances revealing the connection between heat diffusion and architectures of GNNs [8; 51; 7]. Specifically, we interpret feature propagation as heat flows on the underlying Riemmanian manifold, whose characteristics (that are dependent on graph topology and the GNN model) pave the way for a principled representation of the latent geometric knowledge.

## 1.1 Our Contributions

**New theoretical perspective for analyzing latent graph geometry.** On top of the connection between heat equation and GNNs, we step further to inspect the implication of heat kernel for GNNs, and propose a novel notion of *Neural Heat Kernel* (NHK) with rigorous proof of its existence. Heat kernel intrinsically defines the unique solution to the heat equation and can be a fundamental characterization for the geometric property of the underlying manifold [18; 19]. Likewise, NHK uncovers geometric property of the latent graph manifold for GNNs, and governs how information flows between pairs of instances, which lends us a mathematical tool to encapsulate geometric knowledge extracted from GNN model and enables geometric knowledge transfer. This result alone might also be useful in broader contexts for understanding GNNs.

**Flexible distillation framework with versatile instantiations.** Based on the above insights, we treat NHK matrices as representation of the latent geometric knowledge, upon which we build a flexible and principled distillation framework dubbed as *Geometric Knowledge Distillation* (GKD), which aims at encoding and transferring geometric knowledge by aligning latent manifolds behind GNN models as illustrated in Fig. 1. We also develop non-parametirc and parametric versions of GKD, in terms of different ways to approximate NHK computation. Specifically, the former derives explicit NHKs via assumptions on latent space, and the later learns NHK in a data-driven manner.

**Applications for geometric knowledge transfer and conventional KD purposes.** We verify the efficacy of GKD in terms of different geometric knowledge types (i.e., edge-aware and node-aware ones), and further show its effectiveness for conventional KD purposes (e.g., model compression, self distillation, online distillation) for broader applicability. We highlight that our methods consistently exceed teacher model and rival with the oracle model that gives the performance upper bound in principle.

## 1.2 Links to Related Works

**Geometric Deep Learning.** The study of geometric deep learning [5; 38] provides fundamental principles and methodology to generalize deep learning methods to non-Euclidean domains (e.g., graphs and manifolds). From this perspective, architectures for off-the-shelf GNNs [28; 49; 53; 20] and graph Transformers [67; 56] have naturally incorporated the geometric prior knowledge for graphs such as *permutation invariance*. Despite their remarkable success, they highly rely on the graph topology. This work extends the idea of geometric deep learning by treating the global graph

topology as a special type of prior knowledge, and attempts to encode it into GNNs themselves, such that the trained model would leverage information from the global graph topology even without explicitly taking it as input.

**Graph-Based Knowledge Distillation.** Knowledge distillation (KD) [22; 1] uses the outputs of a teacher model as alternative supervised signals to teach a student model, with various new paradigms including feature-based [42; 66] and relation-based [64; 65] ones. While some prior arts [9; 61; 50; 69; 63; 60] attempted to combine KD and GNNs, i.e., graph-based KD, they are nearly straight-forward adaptations of KD without in-depth investigation on the role of graph topology, also restricted by a specific choice of GNN architecture or application scenarios. In contrast, we first formalize the problem of geometric knowledge transfer, theoretically answer the question "how to represent graph geometric knowledge and encode it into GNN models", and propose geometric distillation approach based on the theoretical results. More discussions on related works are deferred to Appendix G.

## 2 Preliminaries

We commence with a brief detour to heat equation on Riemannian manifolds, and its connection with modern GNN architectures. Moreover, we bring forth the notion of *heat kernel* to motivate this work.

**Heat Equation on Manifolds.** We are interested in heat equation defined on a smooth $k$-dimensional Riemannian manifold $\mathcal{M}$. Suppose the manifold is associated with a scalar- or vector-valued function $x(u, t) : \mathcal{M} \times [0, \infty) \to \mathbb{R}^d$, quantifying a specific type of signals such as *heat* at a point $u \in \mathcal{M}$ and time $t$. Fourier's law of heat conductivity describes the flow of heat with respect to time and space, via a partial differential equation (PDE) called *heat equation* [6], i.e.,

$$\frac{\partial x(u, t)}{\partial t} = -c \, \Delta x(u, t), \tag{2}$$

where $c > 0$ is the *thermal conductivity* coefficient, and $\Delta$ is the natural *Laplace–Beltrami operator* associated with $\mathcal{M}$. Rewriting $\Delta$ as the functional composition of the *divergence operator* $\nabla^*$ and *gradient operator* $\nabla$, i.e., $\Delta = \nabla^* \circ \nabla$, we can interpret the heat equation as: the variation of temperature within an infinitesimal time interval at a point is equivalent to the divergence between its own temperature and the average temperature on an infinitesimal sphere around it.

**Implications on Graphs.** A spatial discretisation of a continuous manifold yields a graph $\mathcal{G} = \{\mathcal{V}, \mathcal{E}\}$, whose nodes can be thought of as embedded on the base manifold. In fact, the heat equation along with variants thereof (e.g., Schrödinger equation) have found widespread use in modeling graph dynamics [11; 25; 36]. More importantly, it has been recently revealed to be intimately related with the architectures of modern GNNs [51; 8; 7]: suppose $\mathbf{X}(0) = \{x(u, 0)\}_{u \in \mathcal{V}} \in \mathbb{R}^{n \times d}$ denotes the initial condition for Eqn. (2) determined by input node features, then solving the heat equation under certain definitions of $\nabla^*$ and $\nabla$ (i.e., definition of $\Delta$) amounts to different architectures of GNNs. For instance:

**Example 1.** [51] Define the discretised counterpart of $\Delta$ as the graph Laplacian matrix $\mathbf{L} = \widetilde{\mathbf{D}}^{-\frac{1}{2}}(\widetilde{\mathbf{D}} - \widetilde{\mathbf{A}})\widetilde{\mathbf{D}}^{-\frac{1}{2}}$. Numerically solving Eqn. (2) using the forward Euler method with step size $\tau = 1$ yields the formulation of Simple Graph Convolution (SGC) [53], where $\Theta$ denote learnable transformation matrix

$$\hat{\mathbf{X}}(t) = \left( \tilde{\mathbf{D}}^{-\frac{1}{2}} \tilde{\mathbf{A}} \tilde{\mathbf{D}}^{-\frac{1}{2}} \right)^t \mathbf{X}(0), \quad \hat{\mathbf{Y}} = \mathrm{softmax} \left( \hat{\mathbf{X}}(t) \boldsymbol{\Theta} \right). \tag{3}$$

**Example 2.** [8] Define the gradient operator $\nabla_{ij}$ as the difference of source and target node features, the divergence operator $\nabla_i^*$ as the sum of features of all edges for the node. Numerically solving Eqn. (2) using the explicit Euler scheme with step size $\tau$ yields the following recursive formulation

$$\hat{\mathbf{X}}(t + \tau) = \tau \left( \mathbf{G} - I \right) \hat{\mathbf{X}}(t) + \hat{\mathbf{X}}(t) \tag{4}$$

where $\mathbf{G}$ is a diffusivity coefficient matrix in place of $c$.

Moreover, stacking a non-linear transformation layer after each step yields the formulation of Graph Convolution Networks (GCN) [28] for Eqn. (3), Graph Attention Networks (GAT) [49] with residual connection for Eqn. (4), and even more GNN architectures by virtue of the flexibility of interpretion for heat equation on graphs.

**Heat Kernels.** Intriguingly, it turns out that the initial value problem of heat equation on any manifold $\mathcal{M}$ has a smallest positive fundamental solution depending on the Laplace operator $\Delta$, known as the *heat kernel* [2]. It is denoted as a kernel function $\kappa(x, y, t)$, such that

$$x(u_i, t) = e^{-t\Delta} x(u_i, 0) = \int_{\mathcal{M}} \kappa(u_i, u_j, t) x(u_j, 0) \mathrm{d}\mu(u_j), \tag{5}$$

where $\mu$ is a non-negative measure associated with $\mathcal{M}$. In physics, the heat kernel $\kappa(x, y, t)$ can be interpreted as a transition density that describes the asymptotic behavior of a natural Brownian motion on the manifold. Its formulation thus can be treated as *a unique reflection or representation of the geometry of the underlying manifold*. For example, if the manifold is a $k$-dimensional *Euclidean Space* $\mathbb{R}^k$ or a *Hyperbolic Space* $\mathbb{H}^k$, the explicit formula of heat kernel is respectively given by,

$$\kappa(u_i, u_j, t) = \frac{1}{(4\pi t)^{k/2}} \exp\left(-\frac{\rho^2}{4t}\right) \text{ and } \kappa(u_i, u_j, t) = \frac{(-1)^m}{2^m \pi^m} \frac{1}{(4\pi t)^{\frac{1}{2}}} \left(\frac{1}{\sinh \rho} \frac{\partial}{\partial \rho}\right)^m e^{-m^2 t - \frac{\rho^2}{4t}}, \tag{6}$$

where $\rho = d(u_i, u_j)$ denote geodesic distance. Heat kernel has also been adopted for graph-related applications such as community detection [31], graph clustering [58].

## 3 Extending Heat Kernel to GNNs

The starting point of this work is the development of *neural heat kernel*, built upon the previously-mentioned connection of GNNs and heat equation. As will be discussed later, this novel notion lends us a thermodynamic perspective to the intrinsic geometric property of the latent graph manifold embodied in GNNs, and hence paves the way for distilling geometric knowledge.

### 3.1 Neural Heat Kernel

Consider the graph signal $\mathbf{X}(t)$ at time $t$ and node features $\mathbf{H}^{(l)}$ at layer $l$ as interchangeable notions. Consequently, feature propagation using one layer of GNN amounts to heat diffusion on the base manifold $\mathcal{M}$ within a certain time interval $\tau$, leading to the equivalences of $\mathbf{X}(t + \tau)$ and $\mathbf{H}^{(l+1)}$:

$$\mathbf{H}^{(l+1)} = f_\theta(\mathbf{H}^{(l)}, \mathcal{G}), \ \mathbf{X}(t + \tau) = e^{-\tau \Delta(f_\theta, \mathcal{G})} \mathbf{X}(t), \tag{7}$$

where $f_\theta$ denotes an arbitrary GNN model with parameter $\theta$, and $\Delta(f_\theta, \mathcal{G})$ denotes a generalization of Laplace-Beltrami operator defined over the base manifold $\mathcal{M}$ associated with graph $\mathcal{G}$ and the arbitrary backbone GNN model $f_\theta$.

**Remark.** The equivalence of two equations in Eqn. 7 is based on the recently established connection between heat equation and GNNs [8; 7; 51; 13; 12], which reveal that the formulation of a GNN layer could be thought of as discretisations (that correspond to the left equation in Eqn. 7) of the continuous diffusion process (that correspond to the right equation in Eqn. 7) described by the heat equation. Furthermore, different definitions of Laplace-Beltrami operator $\Delta$ and schemes for solving Eqn. 2 could yield different GNNs (e.g., SGC [53], GAT [49], GRAND [8]). While it is unclear whether there exists such a definition of $\Delta$ for every GNN architecture, we write the operator as $\Delta(f_\theta, \mathcal{G})$ to associate it with model $f_\theta$, and then use the analogy between GNN and heat equation as an analytical tool, in a similar manner with [3; 46; 51; 45; 8; 7], for studying the geometry property of GNNs. See more detailed justifications in Appendix E.

In light of this connection, we consider a natural generalization of heat kernel for GNNs, termed as *neural heat kernel (NHK)* to highlight its difference with heat kernel in the thermodynamic context. In particular, a *single-layer* NHK is defined as a positive definite symmetric kernel function denoted as $\kappa_\theta^{(l)}(v_i, v_j)$, where the sub-script $\theta$ implies that it is associated with the architecture and parameters of the backbone GNN, and the super-script $(l)$ implies that it is specific to each layer, analogous to the role of continuous time $t$ in Eqn. (5).

**Theorem 1.** *(Existence of Single-Layer NHK) Suppose two expressions in Eqn. (7) are equivalent (see Appendix E for more discussions), then for any graph $\mathcal{G}$ and GNN model $f_\theta$, there exist a unique single-layer NHK function $\kappa_\theta^{(l)}(\cdot)$ such that for any node $v_i \in \mathcal{V}$ and $l > 0$,*

$$\mathbf{h}_i^{(l)} = \sum_{v_j \in \mathcal{V}} \kappa_\theta^{(l)}(v_i, v_j) \cdot \mathbf{h}_j^{(l-1)} \mu(v_j) \tag{8}$$

where $\mathbf{h}_i^{(l)} \in \mathbb{R}^d$ denotes the feature of node $v_i$ at $l$-th layer, and $\mu$ is a measure over vertices that could be specified as the inverse of node degree $1/d_i$.

To push further, we can generalize NHK across multiple layers of GNN, termed as a *cross-layer NHK* $\kappa_\theta(v_i, v_j, l \mapsto l+k)$ (e.g., from $l$-th layer to $(l+k)$-th layer of GNN). Its existence could be induced recursively by the *semi-group identity property* of NHK concerning consecutive GNN layers.

**Theorem 2.** *(Semigroup Identity Property of NHK) The NHK satisfies the semigroup identity property: $\forall v_i, v_j \in \mathcal{V}$ and $l > 0$, there exists a cross-layer NHK across two consecutive layers*

$$\kappa_\theta(v_i, v_j, l \mapsto l+2) = \sum_{v_k \in \mathcal{V}} \kappa_\theta^{(l+1)}(v_i, v_k)\kappa_\theta^{(l+2)}(v_k, v_j)d\mu(v_k) \tag{9}$$

This theorem indicates that stacks of multiple GNN layers also constitute a valid kernel, i.e.,

$$\mathbf{h}_i^{(l+k)} = \sum_{v_j \in \mathcal{V}} \kappa_\theta(v_i, v_j, l \mapsto l+k) \cdot \mathbf{h}_j^{(l)} \mu(v_j). \tag{10}$$

Analogous to heat kernel as an unique characterization of the underlying space, NHK characterizes the geometric property of the latent graph manifold for GNNs. Additionally, NHK is dependent on GNN models through the definition of the associated Laplace-Beltrami operator $\Delta(f_\theta, \mathcal{G})$, inheriting the expressiveness of neural networks and varying through the course of training. Intuitively, NHK can be thought of as a model-driven encoding for topological information, encapsulating the geometric knowledge learned by GNNs into a tractable functional form.

## 3.2 Application in Geometric Distillation

Consider the problem of distilling geometric knowledge, which involves an intelligent teacher model $f_{\theta^*}$, which is exposed to and pre-trained over the (relatively) *complete graph* $\tilde{\mathcal{G}} = (\tilde{\mathcal{V}}, \tilde{\mathcal{E}})$, and a student model $f_\theta$ that is exposed to the partial graph $\mathcal{G} = (\mathcal{V}, \mathcal{E})$, where $\mathcal{V} \subseteq \tilde{\mathcal{V}}$ and $\mathcal{E} \subseteq \{\mathcal{V} \times \mathcal{V}\} \cap \tilde{\mathcal{E}}$. Our target is to train a student model (with the help of teacher model) that operates on $\mathcal{G}$ to be as competitive as models operating on $\tilde{\mathcal{G}}$ during inference. Since $\mathcal{G}$ is a sub-graph of $\tilde{\mathcal{G}}$, they should lie in the *same space* (i.e., latent manifold) governed by the underlying mechanism of data generation, and hence we expect student and teacher models to capture the *same geometric property* of this shared space. This leads to the principle of *Geometric Knowledge Distillation* (GKD): transfer the geometric knowledge of the intelligent teacher to the student such that the student can propagate features as if it is aware of the complete graph topology (see the example in Fig. 1).

To this end, we resort to *NHK matrices* on the teacher (resp. student) model over the complete (resp. partial) graph as instantiations of their geometric knowledge, denoted as

$$\text{(Teacher)} \quad \mathbf{K}_{\theta^*}(\tilde{\mathcal{G}}, l \mapsto l+k) = \{\kappa_{\theta^*}(v_i, v_j, l \mapsto l+k)\}_{|\tilde{\mathcal{V}}| \times |\tilde{\mathcal{V}}|},$$

$$\text{(Student)} \quad \mathbf{K}_\theta(\mathcal{G}, l \mapsto l+k) = \{\kappa_\theta(v_i, v_j, l \mapsto l+k)\}_{|\mathcal{V}| \times |\mathcal{V}|},$$

written compactly as $\mathbf{K}^{(l+1)}(\mathcal{G})$ when $k = 1$. The NHK matrix is a positive semi-definite symmetric matrix, and alike $\kappa$, is dependent on the GNN model $f_\theta$ and graph $\mathcal{G}$. Denote $\mathbf{K}_{\theta^*, \mathcal{V}}^{(l)}(\tilde{\mathcal{G}}) \in \mathbb{R}^{|\mathcal{V}| \times |\mathcal{V}|}$ as the sub-matrix of $\mathbf{K}_{\theta^*}^{(l)}(\tilde{\mathcal{G}})$ with row and column indices in $\mathcal{V}$. The distillation loss for GKD is

$$\mathcal{L}_{dis}(\mathbf{K}_{\theta^*, \mathcal{V}}, \mathbf{K}_\theta, l \mapsto l+k) = \mathrm{d}(\mathbf{K}_{\theta^*, \mathcal{V}}(\tilde{\mathcal{G}}, l \mapsto l+k), \mathbf{K}_\theta(\mathcal{G}, l \mapsto l+k)), \tag{11}$$

where $\mathrm{d}(\cdot, \cdot)$ is a similarity measure, for which we choose Frobenius distance as implementation, i.e.,

$$\mathrm{d}(\mathbf{K}_{\theta^*, \mathcal{V}}, \mathbf{K}_\theta) = \|(\mathbf{K}_{\theta^*, \mathcal{V}} - \mathbf{K}_\theta) \odot \mathbf{W}\|_{\mathrm{F}}^2, \quad \mathbf{W}_{v_i, v_j} = \begin{cases} 1 & \text{if} \quad (v_i, v_j) \in \mathcal{E} \\ \delta & \text{if} \quad (v_i, v_j) \notin \mathcal{E}. \end{cases} \tag{12}$$

where $\mathbf{W} \in \mathbb{R}^{|\mathcal{V}| \times |\mathcal{V}|}$ is a weighting matrix to trade-off distillation loss with respect to different node pairs depending on their connectivity. For $k = 1$, the loss can be re-written as $\mathcal{L}_{dis}^{(l+1)}(\mathbf{K}_{\theta^*, \mathcal{V}}, \mathbf{K}_\theta)$. Note that one can also specify different $k$ for teacher and student models in Eqn. (11) in case when the teacher model is deeper.

# 4 Instantiations for Geometric Knowledge Distillation

Unfortunately, deriving explicit formulas for NHKs is prohibitively challenging due to introduction of non-linearity. To circumvent it, we propose two types of instantiations for GKD, i.e., non-parametric and parametric. The former considers explicit NHKs by making assumptions on the underlying space, and the latter learns NHK in a data-driven manner.

## 4.1 Non-Parametric Geometric Distillation

**Deterministic Kernel.** One instantiation of NHK is a *Gauss-Weierstrass kernel* in the form of Eqn. (6), by assuming the underlying space is a Euclidean space. Since the distillation loss in Eqn. (11) is a homogeneous function, we can remove its scaling factor and define NHK as

$$\text{(Gauss-Weierstrass NHK)} \quad \kappa_\theta(v_i, v_j, l \mapsto l+k) \triangleq \exp\left(-\frac{\|\mathbf{h}_i^{(l)} - \mathbf{h}_j^{(l)}\|_2^2}{4T}\right), \quad (13)$$

where $T$ denotes the estimation of the accumulated time interval. Alternatively, we can use *Sigmoid kernel* and define non-parametric NHK as:

$$\text{(Sigmoid NHK)} \quad \kappa_\theta(v_i, v_j, l \mapsto l+k) \triangleq \tanh\left(a \langle \mathbf{h}_i^{(l)}, \mathbf{h}_j^{(l)} \rangle + b\right), \quad (14)$$

where $a, b$ are positive constants depending on $l$ and $k$. It is a natural and intuitive choice as similarity measurement and empirically found as-well effective, albeit does not correspond to any named manifold to our knowledge.

**Randomized Kernel.** We can also define *Randomized kernel* based on the following theorem.

**Theorem 3.** *(Expansion of NHK) Let $\{\varphi_{k'}\}_{k'=0}^{\infty}$ be orthonormal basis of eigenfunctions of $-\Delta(f_\theta, \mathcal{G})$ with eigenvalues $0 < \lambda_0 \leq \lambda_1 \leq \lambda_2 \leq \dots$, NHK allows the expansion:*

$$\kappa_\theta(v_i, v_j, l \mapsto l+k) = \sum_{k'=0}^{\infty} e^{-\lambda_{k'}T} \varphi_{k'}(v_i)^\top \varphi_{k'}(v_j). \quad (15)$$

Based on this result, we resort to the approximation of NHK by defining a randomized kernel in a similar form as Eqn. (15), leading to the following formulation of randomized NHK:

$$\text{(Randomized NHK)} \quad \kappa_\theta(v_i, v_j, l \mapsto l+k) \triangleq \frac{1}{m} \sum_{k'=0}^{m} e^{-\lambda_{k'}T} \left[\sigma\left(\boldsymbol{W}_{k'}\mathbf{h}_i\right)^\top \sigma\left(\boldsymbol{W}_{k'}\mathbf{h}_j\right)\right], \quad (16)$$

where $\sigma\left(\boldsymbol{W}_{k'}\mathbf{h}_i\right)$ is used to proximate $\varphi_{k'}(v_i)$, $\boldsymbol{W}_{k'} = [\boldsymbol{\phi}_{1,k'}, \boldsymbol{\phi}_{2,k'}, \cdots, \boldsymbol{\phi}_{s,k'}]^\top$ is a transformation matrix, $\boldsymbol{\phi} \sim \mathcal{N}\left(\mathbf{0}, \boldsymbol{I}_d\right)$ is a $d$-dimensional random variable from Gaussian distribution. In fact, under certain choice of activation function $\sigma$, Eqn. (16) could approximate a diversity of kernels [41; 10]. This design essentially enforces the alignment of teacher and student for arbitrary underlying manifold.

**Training Scheme.** We follow the standard training paradigm in KD literature [22; 17]: the teacher is pre-trained by a supervised prediction loss involving all labeled nodes in $\tilde{\mathcal{V}}$. After teacher is well-trained, we fix $\theta^*$ and train the student model according to

$$\theta = \arg\min_\theta \mathcal{L}_{pre}(\hat{\mathbf{Y}}_\theta, \mathbf{Y}) + \frac{\alpha}{L} \sum_{l=1}^{L} \mathcal{L}_{dis}^{(l)}(\mathbf{K}_{\theta^*, \mathcal{V}}, \mathbf{K}_\theta), \quad (17)$$

where $\mathbf{Y}$ denotes ground-truth labels of labeled nodes in $\mathcal{V}$, and $\hat{\mathbf{Y}}_\theta$ denotes the predictions of student model $f_\theta$ on $\mathcal{G}$, $\mathcal{L}_{dis}$ is the distillation loss defined by Eqn. (11), $L$ denotes the total number of layers.

## 4.2 Parametric Geometric Distillation

Inheriting the similar spirit of auto-encoding Bayes [27], we introduce a *variational inverse-NHK* that is independently parameterized, denoted as $\kappa_\phi^\dagger$, whose existence is guaranteed by the invertibility of

NHK matrices. Together with $\kappa_\theta$, they define a symmetric form characterizing feature propagation:

$$\text{(Forward)} \quad \mathbf{h}_i^{(l+k)} = \sum_{v_j \in \mathcal{V}} \kappa_\theta(v_i, v_j, l \mapsto l+k) \cdot \mathbf{h}_j^{(l)} \mu(v_j), \tag{18}$$

$$\text{(Backward)} \quad \mathbf{h}_i^{(l)} = \sum_{v_j \in \mathcal{V}} \kappa_\phi^\dagger(v_i, v_j, l+k \mapsto l) \cdot \mathbf{h}_j^{(l+k)} \mu(v_j). \tag{19}$$

In practice, we follow existing kernel learning approaches [52] and parameterize the inverse-NHK as

$$\kappa_\phi^\dagger(v_i, v_j, l+k \mapsto l) = g_\phi(\mathbf{h}_i^{(l+k)})^\top g_\phi(\mathbf{h}_j^{(l+k)}), \tag{20}$$

where $g_\phi : \mathbb{R}^d \to \mathbb{R}^s$ is the associated learnable non-linear mapping. Given a pre-trained teacher model, distilling geometric knowledge boils down to 1) establishing equivalence of Eqn. (18) and Eqn. (19), and 2) matching pseudo-inverse NHK matrices for teacher and student models (respectively denoted as $\mathbf{K}_{\theta^*,\mathcal{V}}^\dagger$ and $\mathbf{K}_\theta^\dagger$ with clear meanings), leading to the training scheme as follows.

**Training Scheme.** Based on Eqn. (19), we can define a *reconstruction loss* with respect to the teacher model (similar applies to the student model) as

$$\mathcal{L}_{rec}(\mathbf{H}_t^{(l+k)}, \mathbf{H}_t^{(l)}) = \|\mathbf{K}_{\theta^*}^\dagger \mathbf{H}_t^{(l+k)} - \mathbf{H}_t^{(l)}\|_F^2. \tag{21}$$

Then, minimizing the reconstruction loss with fixed GNN model parameter $\theta$ amounts to optimizing the variational parameter $\phi$, and minimizing prediction and distillation losses given fixed $\phi$ amounts to optimizing the student model parameter $\theta$:

$$\phi \leftarrow \arg\min_\phi \quad \mathcal{L}_{rec}\left(\mathbf{H}_t^{(l+k)}, \mathbf{H}_t^{(l)}\right) + \mathcal{L}_{rec}\left(\mathbf{H}^{(l+k)}, \mathbf{H}^{(l)}\right), \tag{22}$$

$$\theta \leftarrow \arg\min_\theta \quad \mathcal{L}_{pre}\left(\hat{\mathbf{Y}}_\theta, \mathbf{Y}\right) + \alpha \mathcal{L}_{dis}\left(\mathbf{K}_{\theta^*,\mathcal{V}}^\dagger, \mathbf{K}_{\theta^*,\mathcal{V}}^\dagger, l+k \mapsto l\right). \tag{23}$$

Applying two steps iteratively adds up to an EM-like algorithm for training the student model. In practice, we set $l+k$ as the last layer, and $l$ as the first layer to use as much information as possible. We justify the parametric GKD approach in Appendix. D by showing it essentially explores the true NHK behind GNN.

# 5 Experiments

We conduct experiments to validate the efficacy of our method on graph-structured data in terms of various types of privileged geometric knowledge, combinations of teacher-student GNN architectures and potential application scenarios. We use three benchmark datasets Cora [35], Citeseer [44], Pubmed [39], and a larger dataset OGBN-Arxiv [23] for node classification tasks. More implementation details and experimental results are deferred to Appendix. The codes are available at https://github.com/chr26195/GKD.

**Implementation and Competitors.** We consider the following variants of the proposed GKD. 1) *GKD-G*: non-parametric Gaussian NHK; 2) *GKD-S*: non-parametric Sigmoid NHK; 3) *GKD-R*: randomized NHK; 3) *PGKD*: parametric NHK. We choose KD methods that is representative in its own category for comparison, including *KD* [22], *FSP* [64], *LSP* [63]. We also report the performances of teacher and student model trained with the standard classification loss, short as *Teacher* and *Student*. The teacher model is trained using the complete graph $\tilde{\mathcal{G}}$, and, to calibrate with all other methods, tested using the partial graph $\mathcal{G}$. Besides, we consider an *Oracle* model which is both trained and tested on $\tilde{\mathcal{G}}$, which naturally takes an advantaged place given more information during inference. Since our method is compatible with the vanilla KD paradigm [22], we report the performance delivered by their combinations (i.e., GKD+KD and PGKD+KD).

**Experiment Settings.** We investigate on various experimental settings according to different types of privileged geometric knowledge. In the case of *edge-aware geometric knowledge*, the teacher model has access to additional edge information, i.e., $\mathcal{E} \subset \tilde{\mathcal{E}}$ and $\mathcal{V} = \tilde{\mathcal{V}}$. In the case of *node-aware geometric knowledge*, the teacher model has access to additional node information, i.e., $\mathcal{V} \subset \tilde{\mathcal{V}}$ and $\mathcal{E} = \tilde{\mathcal{E}} \cap \{\mathcal{V} \times \mathcal{V}\}$. We also consider other conventional KD settings including model compression, self-distillation and online distillation, which will be illustrated in detail. The backbone $f_\theta$ is set as 3-layer GCN [28] for both student and teacher models, unless otherwise stated.

| | Cora | CiteSeer | PubMed |
|---|---|---|---|
| Oracle | 88.63 ± 0.48 | 73.64 ± 0.48 | 87.16 ± 0.19 |
| Teacher | 84.61 ± 0.37 | 70.88 ± 0.62 | 84.42 ± 0.52 |
| Student | 83.84 ± 1.32 | 69.94 ± 0.76 | 85.35 ± 0.43 |
| KD | 84.84 ± 1.19 | 70.04 ± 0.37 | 85.58 ± 0.32 |
| FitNets | 83.72 ± 1.45 | 69.99 ± 0.56 | 85.66 ± 0.27 |
| FSP | 83.55 ± 2.19 | 71.43 ± 1.26 | 85.46 ± 0.34 |
| LSP | 83.99 ± 1.39 | 70.23 ± 0.79 | 85.37 ± 0.49 |
| GKD-G | 87.68 ± 1.07 | 73.04 ± 0.70 | 85.74 ± 0.38 |
| GKD-S | 88.01 ± 0.79 | 72.46 ± 0.52 | 85.94 ± 0.43 |
| GKD-R | **88.48 ± 0.59** | 72.97 ± 0.53 | 86.19 ± 0.55 |
| PGKD | 88.41 ± 0.62 | **73.12 ± 0.58** | 86.41 ± 0.24 |
| GKD+KD | 88.95 ± 0.30 | 73.21 ± 0.53 | 86.29 ± 0.28 |
| PGKD+KD | **89.09 ± 0.40** | **73.45 ± 0.48** | **86.48 ± 0.52** |

Table 1: Results of node classification accuracy for the edge-aware knowledge setting.

| | Cora | CiteSeer | PubMed |
|---|---|---|---|
| Oracle | 88.63 ± 0.48 | 73.64 ± 0.48 | 87.16 ± 0.19 |
| Teacher | 87.27 ± 0.51 | 72.92 ± 0.90 | 85.98 ± 0.23 |
| Student | 84.84 ± 1.61 | 70.32 ± 1.12 | 84.74 ± 0.27 |
| KD | 86.71 ± 0.77 | 71.96 ± 1.10 | 85.55 ± 0.45 |
| FitNets | 86.09 ± 1.12 | 72.00 ± 0.78 | 85.78 ± 0.26 |
| FSP | 85.85 ± 1.66 | 70.92 ± 1.46 | 85.20 ± 0.45 |
| LSP | 85.67 ± 1.22 | 70.66 ± 1.01 | 85.71 ± 0.50 |
| GKD-G | 88.66 ± 0.85 | 73.18 ± 0.88 | 86.07 ± 0.45 |
| GKD-S | 88.54 ± 0.52 | 72.85 ± 0.57 | 86.10 ± 0.42 |
| GKD-R | 88.98 ± 0.39 | 72.80 ± 0.22 | **86.16 ± 0.33** |
| PGKD | **89.15 ± 0.45** | **73.33 ± 0.36** | 86.09 ± 0.54 |
| GKD+KD | 89.10 ± 0.44 | 72.94 ± 0.80 | **86.24 ± 0.26** |
| PGKD+KD | **89.23 ± 0.61** | **73.41 ± 0.60** | 86.20 ± 0.36 |

Table 2: Results of node classification accuracy for the node-aware setting.

| Setting | Oracle | Teacher | Student | KD | GKD | PGKD |
|---|---|---|---|---|---|---|
| Edge-Aware | 71.46 ± 0.41 | 67.96 ± 0.78 | 66.41 ± 0.45 | 68.63 ± 1.21 | 70.90 ± 0.80 | **71.38 ± 1.01** |
| Node-Aware | 71.46 ± 0.41 | 69.35 ± 0.72 | 67.49 ± 0.65 | 68.86 ± 0.66 | **71.31 ± 0.83** | 71.27 ± 0.70 |

Table 3: Results of testing accuracy on OGBN-Arxiv dataset.

## 5.1 Main Results

**Edge-Aware Geometric Knowledge.** We report results for the edge-aware geometric knowledge setting. To quantify the privileged information, we set the quantity $(|\tilde{\mathcal{E}}| - |\mathcal{E}|)/|\tilde{\mathcal{E}}|$, called *privileged information ratio* (PIR), as $0.5$. As shown in Tab. 1, all variants of GKD outperform other KD baselines on both datasets, and significantly exceeds both Student and Teacher models. Further, GKD and its variants rival, if not surpass, the Oracle model. In other words, the student model trained using GKD could use far less graph topological information to achieve very close performance to competitors that are aware of the full graph topology during inference. Furthermore, the parametric PGKD performs better than its non-parametric counterparts in most cases, and GKD-R is the most effective non-parametric method in general. Despite that, GKD-G and GKD-S are also effective while being simpler. We presume that the performance variation of different GKD realizations stem from the different geometric property governed by the feature of datasets.

**Node-Aware Geometric Knowledge.** We further investigate on the node-aware geometric knowledge setting where the teacher model has access to more labeled nodes and their relations with the rest nodes. We set the PIR w.r.t. labeled nodes, defined as $(|\tilde{\mathcal{V}}_{train}| - |\mathcal{V}_{train}|)/|\tilde{\mathcal{V}}_{train}|$, to $0.5$. A unique challenge of this setting compared to the edge-ware counterpart is that, apart from graph topological information, the student model has less labeled training samples. As we can see from Tab. 2. The proposed GKD and its variants again consistently outperform KD baselines throughout all the cases, surpasses both Student and Teacher models, and are even as competitive as Oracle.

**Larger Dataset.** Table 3 presents results on a large graph, i.e., OGBN-Arxiv. We use the same PIR setting as citation networks, and choose the best variant of GKD to report in the table. Since the space complexity for GKD is $O(n^2)$, we randomly draw a mini-batch of nodes for computing the distillation loss in practice. Note that the loss function is unbiased as long as all nodes are evenly covered. It could also be seamlessly integrated with the original mini-batch method (by sampling ego-graphs) used for training large graphs without further modification. Again, we found our methods consistently outperform Teacher and Student models, and are close to the performance of the Oracle model, which suggests the effectiveness of GKD in large graphs.

**Performance Variation with Privileged Ratio.** The results with respect to varying privileged (edge-aware and node-ware) information ratio are given respectively in Fig. 2 and Fig. 3. The performance of Oracle model is invariant as it is trained and tested on the same (complete) graph. In general, for Teacher model, Student model, and vanilla KD, their performance drops with increasing PIR quantifying the information loss. In contrast, our method is significantly more robust, only showing slight performance deterioration, exceeding the Teacher model, and approaching the Oracle model. Besides, we find an interesting phenomenon that in the edge-aware setting on Pubmed dataset, the performance of Teacher model is the worst, presumably because that the Teacher is trained using

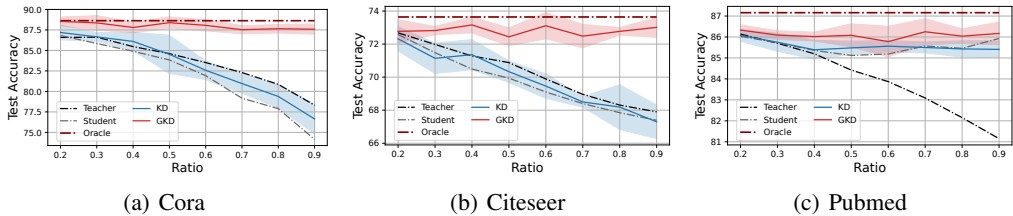

| (a) Cora | (b) Citeseer | (c) Pubmed |

Figure 2: Performance variation with increasing PIR for the edge-aware knowledge setting.

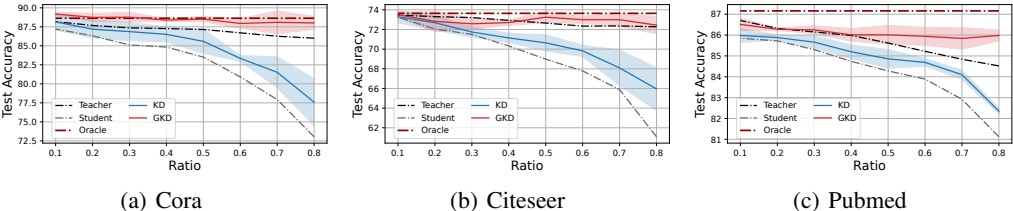

| (a) Cora | (b) Citeseer | (c) Pubmed |

Figure 3: Performance variation with increasing PIR for the node-aware knowledge setting.

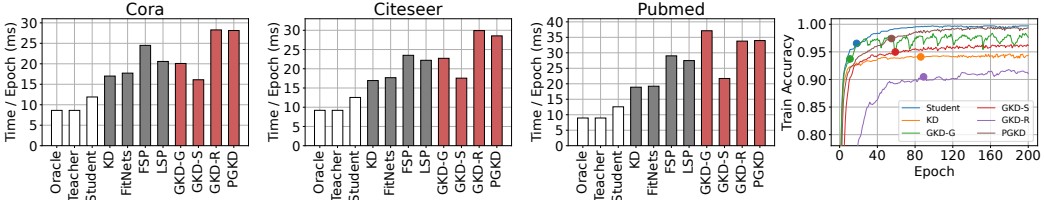

Figure 4: (Left three panels) Training time per epoch (ms) for GKD and baselines on citation networks. (Right panel) Convergence curves of different methods on Cora, where the circle denotes the epoch when the best validation accuracy has been reached.

fully observed graph, and may perform poorly once the privileged part of graph topology becomes unavailable at test time.

## 5.2 Scalability Test

**Training Time.** We report the the training time per epoch of different methods in the left three panels of Fig. 4, where we use the whole graph for training on Cora and Citeseer, and set the batch size as $5,000$ for Pubmed. While the computation complexity for distillation loss of GKD (in both non-parametric and parametric cases) is $O(dn^2)$, we find that in practice their training time is not significantly worse than other KD baselines. In specific, all variants of GKD take less than two times of the training time of vanilla KD while yielding better performance, and the simplest GKD variant using Sigmoid NHK is even more efficient than baselines in some cases. It is also worth mentioning that the distilled student models are equally efficient by using sparser graph structure (in edge-aware setting) or smaller graph (in node-aware setting), which are suitable for being deployed in latency-constrained scenarios.

**Convergence Speed.** We also compare the convergence speed of different variants of GKD and other methods in the right panel of Fig. 4, where we use the same model architecture and optimizer for different methods and finetune other hyper-parameters to ensure fair comparison. Since the loss function is different for different methods, we report the convergence of training accuracy. As shown by the figure, GKD variants using Sigmoid and Gaussian NHKs can converge as fast as the vanilla KD. While GKD-R and PGKD in general take more epochs to converge, they can converge within 200 epochs (which is the default setting recommended in [28]) and the best validation accuracy is achieved within the first 100 epochs. We also find that the EM-style algorithm used for training PGKD does not cost too much extra epochs for convergence than the non-parametric GKD which is reasonable since the learnable mapping $g_\phi$ used in Eqn. (22) is an independent module that is not involved in GNN's feed-forward computation.

|         | Oracle | Teacher | Student | GKD-G | GKD-S | GKD-R | PGKD |
|---------|--------|---------|---------|-------|-------|-------|------|
| Offline | $88.63 \pm 0.48$ | $84.61 \pm 0.37$ | $83.84 \pm 1.32$ | $87.68 \pm 1.07$ | $88.01 \pm 0.79$ | $88.48 \pm 0.59$ | $88.41 \pm 0.62$ |
| Online  | $88.63 \pm 0.48$ | $84.61 \pm 0.37$ | $83.84 \pm 1.32$ | $87.75 \pm 0.65$ | $87.63 \pm 0.65$ | $88.28 \pm 0.80$ | $88.50 \pm 0.33$ |

Table 4: Comparison of offline and online distillation on Cora in edge-aware setting.

## 5.3 Other Settings

| Setting | Teacher | Student | KD | GKD |
|---------|---------|---------|-----|-----|
| Compression | | SGC $85.93 \pm 0.17$ | SGC $86.32 \pm 0.32$ | SGC $87.15 \pm 0.85$ |
| Compression | GCN-64 $88.76 \pm 0.34$ | GCN-8 $84.52 \pm 1.34$ | GCN-8 $87.50 \pm 1.04$ | GCN-8 $88.30 \pm 0.46$ |
| Compression | | GCN-16 $87.24 \pm 0.43$ | GCN-16 $88.09 \pm 0.79$ | GCN-16 $88.62 \pm 0.50$ |
| Self-Distil | GCN-32 $88.63 \pm 0.48$ | GCN-32 $88.98 \pm 0.34$ | | GCN-32 $89.23 \pm 0.52$ |
| Self-Distil | GCN-16 $87.24 \pm 0.43$ | GCN-16 $87.62 \pm 0.52$ | | GCN-16 $88.56 \pm 0.40$ |

Table 5: Node classification accuracy on Cora in settings: 1) model compression; 2) self-distillation.

**Model Compression.** We report the performance of GKD in the conventional model compression setting [22] where the teacher and student are different-sized GNN models. In this setting, we train and test both teacher and student models on the complete graph. We use GCN as the backbone for teacher with a relatively large hidden size $64$, and use SGC or GCN (with hidden size 8 and 16) as the student model. We report the best result among all variants of GKD. As shown in the first section of Tab. 5, our method achieves notable improvements over the student model, rendering it as a useful KD approach for model compression.

**Self-Distillation.** We further report results for the setting of *self-distillation* [15; 37], which is a special case of KD when the teacher's and student's architectures are identical, often used for refining their own performance. The results as shown in Tab. 5 validate that GKD could also be used to effectively boost GNN's own performance.

**Online Distillation.** Table. 4 shows the performance of different variants of GKD for online distillation where both the teacher model and the student model are trained in an end-to-end manner, in contrast to offline distillation where the teacher model is pre-trained. The results demonstrate the potential usage of GKD in this setting.

## 6 Conclusion, Current Limitations and Future Works

This paper formalizes the problem of graph topological knowledge transfer for GNNs. We investigate on the implication of heat kernel in GNNs and propose the novel notion of neural heat kernel. We leverage it to characterize the geometric property of the underlying manifold for graphs, and propose the framework of geometric knowledge distillation to transfer geometric knowledge from a teacher GNN model to a student GNN. Experimental results validate the effectiveness of our approach in various practical settings.

Despite that the proposed GKD is effective in various tasks and possesses decent training efficiency in practice, its theoretical space and time complexities are $O(n^2)$ and $O(dn^2)$ and the parametric instantiation may take some extra time to converge compared to the pure non-KD counterpart. The algorithmic complexity can be reduced by using mini-batch training, and there also exist ways to reduce the overhead such as pre-computing teacher's NHK matrix or using low-rank approximation. Finally, we do not foresee any direct negative societal impacts of this work.

## Acknowledgement

This work was partly supported by National Key Research and Development Program of China (2020AAA0107600), National Natural Science Foundation of China (61972250, 72061127003), and Shanghai Municipal Science and Technology (Major) Project (22511105100, 2021SHZDZX0102).

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
