# A  Proof for Theorem 1

Suppose $(\mathcal{M}, \mu)$ is the base manifold of dimension $n$ with respect to graph $\mathcal{G}$ and GNN model $f_\theta$, and $\Delta_\mu(\mathcal{G}, f_\theta)$ is the associated weighted Laplace operator with respect to $\mathcal{G}$ and $f_\theta$ (in the following shortened as $\Delta_\mu$). We leverage a Sobolev inequality on manifolds as Lemma 1.

**Lemma 1.** *[43] Let $x$ be a function from the local Sobolev space $\mathcal{W}_{loc}^{2\sigma}(\mathcal{M})$ for a positive integer $\sigma > n/4$. Then, for any relatively compact open set $\Omega \subset \mathcal{M}$ and any set $K \Subset \Omega$, there is a constant $C$ such that*

$$\sup_K |x| \leq C \|x\|_{\mathcal{W}^{2\sigma}(\Omega)}. \tag{24}$$

*where the norm $\|\cdot\|_{\mathcal{W}^{2\sigma}}$ is defined as*

$$\|x\|_{\mathcal{W}^{2\sigma}}^2 = \sum_{l=0}^{k} \left\|\Delta_\mu^l x\right\|_{L^2}^2 \tag{25}$$

Suppose $\sigma$ is the smallest integer larger than $n/4$, and $\mathbf{P}_t = e^{-t\mathcal{L}}$ is the heat kernel semigroup, where $\mathcal{L} = -\Delta_\mu|_{W_0^2}$ is the Dirichlet Laplace operator for the base manifold regarding graph $\mathcal{G}$ and GNN model $f_\theta$, we have the following lemma.

**Lemma 2.** *For any function $x \in L^2(\mathcal{M})$, $t > 0$, and set $K \Subset \mathcal{M}$, it holds that*

$$\sup_K |\mathbf{P}_t x| \leq C \left(1 + t^{-\sigma}\right) \|x\|_{L^2(\mathcal{M})}, \tag{26}$$

*where $C$ is a constant depending on $K, \mathbf{g}, \mu, n$.*

*Proof.* Suppose $\{E_\lambda\}$ is the spectral resolution of the Dirichlet operator $\mathcal{L}$ for the base manifold. Consider the function $\Phi(\lambda) = \lambda^k e^{-t\lambda}$, where $t > 0$ and $k \in \mathbb{Z}^+$. We have

$$\mathcal{L}^k e^{-t\mathcal{L}} = \int_0^\infty \Phi(\lambda) dE_\lambda \tag{27}$$

Since $\Phi(\lambda)$ is bounded on $[0, +\infty)$, $\mathcal{L}^k e^{-t\mathcal{L}}$ is also bounded, and hence we have $\mathcal{L}^k \left(e^{-t\mathcal{L}} x\right) \in L^2(\mathcal{M})$. By noticing that the function $\lambda \mapsto \lambda^k e^{-t\lambda}$ takes its maximal value at $\lambda = k/t$, we have, for any $x \in L^2$,

$$\begin{aligned}
\left\|\Delta_\mu^k \mathbf{P}_t x\right\|_{L^2} &= \left\|\mathcal{L}^k e^{-t\mathcal{L}} x\right\|_{L^2} \\
&= \left(\int_0^\infty \left(\lambda^k e^{-t\lambda}\right)^2 d\|E_\lambda x\|_{L^2}^2\right)^{1/2} \\
&\leq \sup_{\lambda \geq 0} \left(\lambda^k e^{-t\lambda}\right) \left(\int_0^\infty d\|E_\lambda x\|_{L^2}^2\right)^{1/2} \\
&= \left(\frac{k}{t}\right)^k e^{-k} \|x\|_{L^2}.
\end{aligned} \tag{28}$$

Using the definition of $\|\cdot\|_{\mathcal{W}^{2\sigma}}$ in Eqn. (25) and the result of Eqn. (28), we obtain

$$\begin{aligned}
\|\mathbf{P}_t x\|_{\mathcal{W}^{2\sigma}} &= \sum_{k=0}^{\sigma} \left(\frac{k}{t}\right)^k e^{-k} \|x\|_{L^2} \\
&\leq C \left(1 + \sum_{k=1}^{\sigma} \left(\frac{k}{t}\right)^k e^{-k}\right) \|x\|_{L^2} \\
&\leq C' \left(1 + t^{-\sigma}\right) \|x\|_{L^2}.
\end{aligned} \tag{29}$$

Substituting $\mathbf{P}_t x$ into Lemma. 1 and using the result of Eqn. 29 yields Eqn. 26, and thus completes the proof. $\square$

Consider the following equivalent expressions describing heat diffusion and feature propagation respectively,

$$\mathbf{H}^{(l+1)} = f_\theta(\mathbf{H}^{(l)}, \mathcal{G}), \quad \mathbf{X}(t + \tau) = e^{-\tau \Delta(f_\theta, \mathcal{G})} \mathbf{X}(t) \tag{30}$$

where $\tau$ is a constant dependent on $f_\theta$ and $l$. We can rewrite the single-layer NHK as a function $p_{\tau, v_i}$ such that for any $v_j \in \mathcal{V}$ (or equivalently $v_j \in \mathcal{M}$)

$$p_{\tau, v_i}(v_j) = \kappa_\theta^{(l)}(v_i, v_j). \tag{31}$$

We define $g : \mathcal{V} \to \mathbb{R}^d$ as a function that outputs the feature of a node in $l$-th layer, which is clearly Lebesgue integrable by thinking of nodes embedded on the manifold, i.e., $g \in L^2(\mathcal{M})$.

Proof for the existence of unique single-layer NHK boils down to proof that for any $v_i \in \mathcal{M}, l > 0$ (i.e., $\tau > 0$), there exists a unique function $p_{\tau, v_i} \in L^2(\mathcal{M})$ such that, for all $g_\theta$,

$$\mathbf{P}_\tau g_\theta(v_i) = \int_{v_j} p_{\tau, v_i} g_\theta(v_j) \mathrm{d}v_j. \tag{32}$$

Fix a relatively compact set $K \Subset \mathcal{M}$. By Lemma 2, for all $\tau > 0$ and $g_\theta \in L^2(\mathcal{M})$, the function $\mathbf{P}_\tau g_\theta(v_i)$ admits the estimate

$$|\mathbf{P}_\tau g_\theta(v_i)| \leq C \left(1 + \tau^{-\sigma}\right) \|g_\theta(v_i)\|_{L^2(\mathcal{M})}. \tag{33}$$

Therefore, for fixed $l$ and GNN model $f_\theta$, the mapping $g_\theta \mapsto \mathbf{P}_\tau g_\theta$ is a bounded linear functional on $L^2(\mathcal{M})$. By the Riesz representation theorem, there exists a function $p_{\tau, v_i} \in L^2(\mathcal{M})$ such that

$$\mathbf{P}_\tau g_\theta = (p_{\tau, v_i}, g_\theta)_{L^2} \quad \text{for all } g_\theta \in L^2(\mathcal{M}), \tag{34}$$

where $(\cdot, \cdot)_{L^2}$ denotes inner product in $L^2$, whence Eqn. (32) follows. The uniqueness of $p_{\tau, v_i}$ is evident from Eqn. (32) since for any point $v_i \in \mathcal{M}$ there is a compact set $K$ containing $v_i$, the function $p_{\tau, v}$ is defined for all $\tau > 0$ and $v \in \mathcal{M}$.

# B  Proof for Theorem 2

**Lemma 3.** *For all $v_i, v_j \in \mathcal{M}$ and $t > 0$, the inner product $(p_{s,x}, p_{t-s,y})$ does not depend on $s \in (0, t]$.*

*Proof.* Using $\mathbf{P}_{t+s} = \mathbf{P}_s \mathbf{P}_t$ (by the definition of $\mathbf{P}_t$), Eqn. (34), and the symmetry of $\mathbf{P}_t$, we obtain that for all $v \in \mathcal{M}, t, s > 0$, and $g_\theta \in L^2(\mathcal{M})$, it holds that

$$\begin{aligned}
\mathbf{P}_{t+s} g_\theta(v_i) &= \mathbf{P}_s \left(\mathbf{P}_t g_\theta\right)(v_i) \\
&= (\mathbf{P}_{s, v_i}, \mathbf{P}_t g_\theta) = (\mathbf{P}_t \mathbf{P}_{s, v_i}, g_\theta) \\
&= \int_{\mathcal{M}} \mathbf{P}_t \mathbf{P}_{s, v_i}(z) g_\theta(v_j) \mathrm{d}\mu(v_j) \\
&= \int_{\mathcal{M}} \left(\mathbf{P}_{t, v_j}, \mathbf{P}_{s, v_i}\right) g_\theta(v_j) \mathrm{d}\mu(v_j),
\end{aligned} \tag{35}$$

$\square$

Lemma 3 implies that, for all $v_i, v_j \in \mathcal{M}$ and $0 < s \leq t$

$$p_t(v_i, v_k) = \left(p_{s, v_i}, p_{t-s, v_k}\right). \tag{36}$$

Hence, it holds that

$$\begin{aligned}
\int_{\mathcal{M}} p_t(v_i, v_j) p_s(v_j, v_k) d\mu(v_j) \\
= (p_t(v_i, \cdot), p_s(v_k, \cdot)) = p_{t+s}(v_i, v_k).
\end{aligned} \tag{37}$$

By letting

$$s = \min\{\tau^{(l+1)}, \tau^{(l+2)}\}, \quad t = \max\{\tau^{(l+1)}, \tau^{(l+2)}\}, \tag{38}$$

we can rewrite Eqn. (37) as the expression of semigroup identity property of layer-wise NHK

$$\begin{aligned}
\kappa_\theta(v_i, v_j, l \mapsto l+2) = \\
\sum_{v_k \in \mathcal{V}} \kappa_\theta^{(l+1)}(v_i, v_k) \kappa_\theta^{(l+2)}(v_k, v_j) d\mu(v_k).
\end{aligned} \tag{39}$$

The proof also generalizes to the cross-layer case, inducing cross-layer NHK $\kappa_\theta(v_i, v_j, l \mapsto l+k)$.

## C  Proof for Theorem 3

Let $\{\varphi_k\}_{k=1}^{\infty}$ be an orthonormal basis of eigenfunctions of $\mathcal{L}$ with an increasing sequence of non-negative eigenvalues $\{\lambda_k\}_{k=1}^{\infty}$, where $\lambda_k \to +\infty$. In consideration of

$$(p_{\tau,v_i}, \varphi_k)_{L^2} = P_\tau \varphi_k(v_i) = e^{-\tau\mathcal{L}}\varphi_k(v_i) = e^{-\tau\lambda_k}\varphi_k(v_i), \tag{40}$$

we have the following expansion of $p_{t,v_i}$ by referring to results in literature [48]

$$p_{t,v_i} = \sum_k e^{-t\lambda_k}\varphi_k(v_i)\varphi_k. \tag{41}$$

Let $T$ be the accumulated time interval from $l$-th layer to $(l+k)$-th layer, in consideration of the equivalence shown in Eqn. (31), we could write Eqn. (41) as

$$\kappa_\theta(v_i, v_j, l \mapsto l+k) = \sum_{k'=0}^{\infty} e^{-\lambda_{k'}T}\varphi_{k'}(v_i)^\top \varphi_{k'}(v_j) \tag{42}$$

completing the proof.

## D  Justification for Parametric GKD

We justify parametric GKD from a variational inference perspective. From Eqn. (18), the forward GNN model $f_\theta$ defines a model distribution

$$p_\theta(\mathbf{H}^{(l)}, \mathbf{H}^{(l+k)}, \mathbf{K}) = p_\theta(\mathbf{H}^{(l)})p_\theta(\mathbf{K}|\mathbf{H}^{(l)})p_\theta(\mathbf{H}^{(l+k)}|\mathbf{K}, \mathbf{H}^{(l)}), \tag{43}$$

where $p_\theta(\mathbf{K}|\mathbf{H}^{(l)})$ is intractable, hindering the proceeding distillation. In this light, the variational inverse-NHK model $\kappa_\phi^\dagger$ is proposed with a variational distribution

$$q_\phi(\mathbf{H}^{(l)}, \mathbf{H}^{(l+k)}, \mathbf{K}) = q_\phi(\mathbf{H}^{(l+k)})q_\phi(\mathbf{K}|\mathbf{H}^{(l+k)})q_\phi(\mathbf{H}^{(l)}|\mathbf{H}^{(l+k)}, \mathbf{K}), \tag{44}$$

which has a tractable posterior $q_\phi(\mathbf{K}|\mathbf{H}^{(l+k)})$. Now, we justify our training scheme with iterative optimization for Eqn. (22) and (23) by the following proposition.

**Proposition 1.** *The optimization in Eqn. (22) and (23) essentially minimizes the following Kullback–Leibler (KL) divergence,*

$$\min_{\theta,\phi} \mathcal{D}_{kl}\left(q_\phi(\mathbf{K}, \mathbf{H}^{(l)}, \mathbf{H}^{(l+k)}) \,\|\, p_\theta(\mathbf{K}, \mathbf{H}^{(l)}, \mathbf{H}^{(l+k)})\right), \tag{45}$$

*and hence attempts to establish equivalence between two latent variable models $p_\theta$ and $q_\phi$.*

*Proof.* In the following, we use $\mathbf{X}, \mathbf{Y}, \mathbf{K}$ to denote $\mathbf{H}^{(l)}, \mathbf{H}^{(l+k)}, \mathbf{K}(\mathcal{G}, l \mapsto l+k)$ for brevity. By definition in Section 4.2 we have a forward GNN model $f_\theta$ with the joint distribution of latent variables

$$p_\theta(\mathbf{X}, \mathbf{Y}, \mathbf{K}) = p_\theta(\mathbf{X})\underbrace{p_\theta(\mathbf{K}|\mathbf{X})}_{\text{Intractable}}p_\theta(\mathbf{Y}|\mathbf{X}, \mathbf{K}), \tag{46}$$

and a variational inverse-NHK model $\kappa_\phi^\dagger$ with the joint distribution of latent variables

$$q_\phi(\mathbf{X}, \mathbf{Y}, \mathbf{K}) = q_\phi(\mathbf{Y})\underbrace{q_\phi(\mathbf{K}|\mathbf{Y})}_{\text{Tractable}}q_\phi(\mathbf{X}|\mathbf{Y}, \mathbf{K}). \tag{47}$$

Based on these equations, we can write the KL-divergence between joint distributions of $p_\theta$ and $q_\phi$ as

$$
\begin{aligned}
&\mathcal{D}_{\mathrm{KL}}(q_\phi(\mathbf{X}, \mathbf{Y}, \mathbf{K}) \parallel p_\theta(\mathbf{X}, \mathbf{Y}, \mathbf{K})) \\
&= \iiint q_\phi(\mathbf{X}, \mathbf{Y}, \mathbf{K}) \log \frac{q_\phi(\mathbf{X}, \mathbf{Y}, \mathbf{K})}{p_\theta(\mathbf{X}, \mathbf{Y}, \mathbf{K})} \mathrm{d}\mathbf{X}\mathrm{d}\mathbf{Y}\mathrm{d}\mathbf{K} \\
&= \mathbb{E}_{q_\phi(\mathbf{Y})}\left[\iint q_\phi(\mathbf{K}, \mathbf{X}|\mathbf{Y}) \log \frac{q_\phi(\mathbf{Y})q_\phi(\mathbf{K}, \mathbf{X}|\mathbf{Y})}{p_\theta(\mathbf{X}, \mathbf{Y}, \mathbf{K})} \mathrm{d}\mathbf{X}\mathrm{d}\mathbf{K}\right] \\
&= C' + \mathbb{E}_{q_\phi(\mathbf{Y})}\left[\iint q_\phi(\mathbf{K}, \mathbf{X}|\mathbf{Y}) \log \frac{q_\phi(\mathbf{K}, \mathbf{X}|\mathbf{Y})}{p_\theta(\mathbf{X}, \mathbf{Y}, \mathbf{K})} \mathrm{d}\mathbf{X}\mathrm{d}\mathbf{K}\right] \\
&= C' + \mathbb{E}_{q_\phi(\mathbf{Y})}\left[\mathbb{E}_{q_\phi(\mathbf{K}|\mathbf{Y})}\left[\int_{\mathbf{X}} q_\phi(\mathbf{X}|\mathbf{Y}, \mathbf{K})\cdot \right.\right. \\
&\qquad\qquad \left.\left. \log \frac{q_\phi(\mathbf{K}|\mathbf{Y})q_\phi(\mathbf{X}|\mathbf{Y}, \mathbf{K})}{p_\theta(\mathbf{X}, \mathbf{Y}, \mathbf{K})} \mathrm{d}\mathbf{X}\right]\right] \\
&= C + \mathbb{E}_{q_\phi(\mathbf{Y})}\left[\mathbb{E}_{q_\phi(\mathbf{K}|\mathbf{Y})}\left[\int_{\mathbf{X}} q_\phi(\mathbf{X}|\mathbf{Y}, \mathbf{K})\cdot \right.\right. \\
&\qquad\qquad \left.\left. \log \frac{q_\phi(\mathbf{X}|\mathbf{Y}, \mathbf{K})}{p_\theta(\mathbf{X})p_\theta(\mathbf{Y}, \mathbf{K}|\mathbf{X})} \mathrm{d}\mathbf{X}\right]\right] \\
&= C + \mathbb{E}_{q_\phi(\mathbf{Y})}\left[\mathbb{E}_{q_\phi(\mathbf{K}|\mathbf{Y})}\left[\underbrace{\mathcal{D}_{\mathrm{KL}}(q_\phi(\mathbf{X}|\mathbf{Y}, \mathbf{K}) \parallel p_\theta(\mathbf{X}))}_{\text{Reconstruction Loss}} - \underbrace{\mathbb{E}_{q_\phi(\mathbf{X}|\mathbf{Y}, \mathbf{K})}[\log p_\theta(\mathbf{K}, \mathbf{Y}|\mathbf{X})]}_{\text{Prediction Loss}}\right]\right].
\end{aligned}
\tag{48}
$$

The first term $C$ is a constant entropy with respect to $q_\phi$. The second term is the KL-divergence between the variational posterior $q_\phi(\mathbf{X}|\mathbf{Y}, \mathbf{K})$ and the prior $p_\theta(\mathbf{X})$, which corresponds to the reconstruction loss in Eqn. (22) that attempts to learn a NHK that faithfully reflects the latent heat diffusion process describing the GNN feature propagation. The third term is negative log-likelihood, which corresponds to the prediction loss in Eqn. (23) that attempts to fit the dataset. Therefore, minimizing this KL-divergence amounts to the iterative optimization scheme of Eqn. (22) and (23). □

## E    More Discussions on the Equivalence of Eqn.7

The equivalence of two equations in Eqn. 7 is based on recent works [8; 7; 51; 13; 12] that built connection between heat equation and GNN. The main result of these works is that by treating node features H as signal X (corresponding to $x(u, t)$ in heat equation Eqn. 2) on the graph, solving the heat equation with Euler scheme yields the formulation of a GNN layer. In other words, the GNN can be seen as the discretisations of the continuous diffusion process described by the heat equation. Correspondingly in Eqn. 7, the left equation is a general GNN layer (corresponding to discretized diffusion process), and the right equation is directly derived from Eqn. 7 (corresponding to continuous diffusion process).

Moreover, different definitions of Laplace-Beltrami operator $\Delta$ yield different GNNs. To be more specific, the simplest definition [8; 51] of $\Delta$ is by letting the gradient operator $\nabla$ denote assigning each edge the difference of adjacent nodes' features, and the divergence operator $\nabla^*$ denote the summation of edge features obtained from last step. In this case, setting $\tau$ in Eqn. 2 as 1 yields SGC [53], and since it is being pointed out to be a potential cause of over-smoothing (which is one example to show the thermodynamic and geometric perspective is helpful for understanding GNNs), the authors in [51] further set smaller $\tau$ which yields DGC. Reinterpreting $\nabla^*$ as weighted sum of edge features yields GAT [8; 49] and any other GNN following the message passing scheme:

$$
\mathrm{MessagePassing}(\mathbf{h}_u, \mathcal{G}) = \sum_{v \in \mathcal{N}_u \cup u} s(\mathbf{h}_u, \mathbf{h}_v) \cdot \mathbf{h}_v,
\tag{49}
$$

where $\mathcal{N}_u$ denotes neighbored nodes of $u$, and $s$ denotes a parametric or non-parametric similarity score. The neighborhood summation could also be modified by changing the definition of $\nabla$, which yields GNNs based on learned structures (e.g., GDC [30]) and those with residual links (e.g., APPNP [29]). Further considering different discretization schemes (e.g., implicit scheme, multi-step schemes) yields different variants of GRAND [8; 7; 45] that are found more powerful than its simplification (presumably because it better matches the continuous diffusion process).

Unfortunately, not all GNNs have a simple form of $\Delta$, and for some of them, whether there exists such $\Delta$ is an open research question. Therefore, we write the operator as $\Delta(f_\theta, \mathcal{G})$ to associate it with model $f_\theta$ and use equivalence in Eqn. 2 as an analytical assumption. The thermodynamic and geometric perspective used in the paper is fundamental and useful for studying the geometry property of GNNs, which has also been adopted by other works [3; 46; 51; 45; 8; 7]. For example, [3] attempts to explain heterophily and oversmoothing problems in GNNs by connecting GNNs to the heat equation defined by the sheaf Laplace-Beltrami operator $\Delta_\mathcal{F}$ and propose new GNN architectures based on the theory. Similarly, [45; 8; 7] propose new GNN architectures / rewiring methods from the same thermodynamic perspective, [51] explains over-smoothing issue of SGC based on the same equivalence of GNN and solve of heat equation, and [46] explains the over-squashing issue from a geometric perspective by analyzing the curvature. A common trait of these works is to draw analogies between GNNs and differential geometry / diffusion process to obtain meaningful theoretical results (ours: NHK as a characterization of GNN's underlying geometry) that are used to guide implementation (ours: GKD for geometric knowledge transfer).

## F    Implementation Details

We present implementation details for our experiments for reproducibility. We implement our model as well as the baselines with Python 3.7, Pytorch 1.8 and Pytorch Geometric 1.7. All experiments are conducted on NVIDIA V100 with 16 GB memory. All parameters are initialized with Xavier initialization procedure. We train the model by Adam optimizer. Both teacher and student models are trained from scratch. For the main results reported in Tab. 1 and 2, we choose the backbone model as a 3-layer GCN with hidden size 32. In case that the graph is too large, we use mini-batch training (draw a mini-batch of nodes from the vertex set $\mathcal{V}$) for computing the distillation loss. All the experiments are repeated five times with random initialization.

### F.1    Dataset Description

We choose three benchmark citation network datasets, i.e., Cora, Citeseer and Pubmed, and a large-scale network dataset OGB-Arxivfor node classification. For citation networks, we randomly split them into train/valid/test data according to the ratio 2:1:1. For OGB dataset, we follow the original splitting [23] for evaluation. The statistics of these datasets are summarized in Tab. 6.

| Dataset | # Classes | # Nodes | # Edges | Metric |
|---------|-----------|---------|---------|--------|
| Cora [35] | 7 | 2,485 | 5,069 | Accuracy |
| Citeseer [44] | 6 | 2,120 | 3,679 | Accuracy |
| PubMed [39] | 3 | 19,717 | 44,324 | Accuracy |
| OGB-Arxiv [23] | 40 | 169,343 | 1,166,243 | Accuracy |

Table 6: Statistics of Datasets.

### F.2    Hyper-Parameter Tuning

For parameter tuning, we adopt grid search method to search for hyper-parameters on validation set. Since our model is only sensitive to $\alpha$, one can use other more efficient searching strategies instead of grid search (e.g., coordinate descent, Bayesian optimization) to achieve very similar results. The descriptions for several hyper-parameters and their search spaces are shown in Tab. 7.

| Notation | Description | Search Space |
|----------|-------------|--------------|
| $T$ | accumulated time interval | [0.25, 0.5, 1, 2, 4] |
| $\alpha$ | weight of the geometric distillation loss | [0.1, 0.3, 1.0, 3.0, 10.0, 30.0, 100.0, 300.0, 1000.0] |
| $\alpha_{kd}$ | weight of the label-based distillation loss | [0.0, 0.2, 0.4, 0.6, 0.8] |
| $\tau$ | temperature for label-based distillation loss | [0.25, 0.5, 1, 2, 4] |
| $\delta$ | weight for non-connection entries for distillation loss | [0.0, 0.1, 0.2, 0.4, 0.6, 0.8, 1, 2] |
| $\gamma$ | learning rate | [0.0001, 0.001, 0.01, 0.1] |

Table 7: Parameter searching space for GKD and its variants.

### F.3 Implementation of non-parametric GKD

For Gauss-Weierstrass NHK in Eqn. (13), we treat the accumulated time interval $T$ as a hyper-parameter that is consistent for all layers. For sigmoid NHK in Eqn. (13), we let $a = 1$, $b = 0$ and the NHK is simple dot-product with activation. For randomized NHK in Eqn. (16), we let $s = 2d$ for the random transformation matrix $\mathbf{W}$, and we use $tanh(\cdot)$ as activation function for $\sigma(\cdot)$.

---

**Algorithm 1:** Training Algorithm for GKD.

---

**Input:** Complete graph $\tilde{\mathcal{G}}$ with labels $\tilde{\mathbf{Y}}$, partial graph $\mathcal{G}$ with labels $\mathbf{Y}$, learning rate $\gamma$, initial parameters $\theta^*$, $\theta$.

**Training Teacher GNN**:

**while** *Not converged* **do**

    Teacher model conducts feature propagation on $\tilde{\mathcal{G}}$

    Calculate $\mathcal{L}_{pre}(\hat{\mathbf{Y}}_{\theta^*}, \tilde{\mathbf{Y}})$ as in Eqn. (17)

    $\theta^* \leftarrow \theta^* - \gamma \nabla_{\theta^*} \mathcal{L}_{pre}(\hat{\mathbf{Y}}_\theta, \tilde{\mathbf{Y}})$

Save teacher model as $f_{\theta^*}$

**Training Student GNN**:

Load teacher model $f_{\theta^*}$

**while** *Not converged* **do**

    Teacher model conducts feature propagation on $\tilde{\mathcal{G}}$

    Student model conducts feature propagation on $\mathcal{G}$

    Calculate $\mathcal{L}_{pre}$ and $\mathcal{L}_{dis}$ as in Eqn. (17)

    $\theta \leftarrow \theta - \gamma \nabla_\theta \mathcal{L}_{pre}(\hat{\mathbf{Y}}_\theta, \mathbf{Y}) + \frac{\alpha}{L} \sum_{l=1}^{L} \mathcal{L}_{dis}^{(l)}$

---

### F.4 Implementation of parametric GKD

For the case of parametric GKD, we realize the non-linear mapping $g_\phi$ as a neural network one-layer feed-forward neural network with $tanh(\cdot)$ activation, and set $s = 2d$, the same as the non-parametric setting. In case that the teacher model has larger hidden size than the student model, we follow standard approaches in feature-based knowledge distillation methods [21] and use independent mappings for teacher and student models that are customized for their own hidden sizes. We use an EM-style algorithm for training the student model as in Eqn. (22) and Eqn. (23). For variants of GKD+KD and PGKD+KD, we consider additional standard label-based distillation loss in [22] with respect to labeled nodes, inducing an additional hyper-parameter $\alpha_{kd}$ that controls its importance.

### F.5 Descriptions and Implementations of Baselines

**KD** [22]: is the seminal work of knowledge distillation, which uses the predictions of the teacher model as soft labels to teach the student model.

**FitNets** [42]: is the representative work of feature-based knowledge distillation, using the intermediate layers of training instances to teach the student model.

**FSP** [64]: is a relation-based knowledge distillation method, using the Gram matrix between two intermediate layers to explore the relationships between different feature maps.

**LSP** [63]: is a knowledge distillation method for graph convolutional networks, which aims to match the local distribution (similarity with adjacent nodes) of teacher and student models.

Note that some of these baselines (FitNets and FSP) are originally designed specifically for computer vision tasks, we adapt them to the setting of graph neural networks with slight modifications. While FSP is designed for graph neural networks, its original formulation of distillation loss does not account for the difference of graph topology for teacher and student models. To make it compatible with the settings considered in this paper, we fix the definition of local structures with respect to either student or teacher graph topology, and choose the best result for report. We refer to the hyper-parameter settings in their papers and also finetune them on different datasets.

**Algorithm 2:** Training Algorithm for PGKD.

---

**Input:** Complete graph $\tilde{\mathcal{G}}$ with labels $\tilde{\mathbf{Y}}$, partial graph $\mathcal{G}$ with labels $\mathbf{Y}$, learning rate $\gamma_1$ and $\gamma_2$, initial parameters $\theta^*$, $\theta$, $\phi$.

**Training Teacher GNN**:

**while** *Not converged* **do**

 | Teacher model conducts feature propagation on $\tilde{\mathcal{G}}$
 | Calculate $\mathcal{L}_{pre}(\hat{\mathbf{Y}}_{\theta^*}, \tilde{\mathbf{Y}})$ as in Eqn. (17)
 | $\theta^* \leftarrow \theta^* - \gamma_1 \nabla_{\theta^*} \mathcal{L}_{pre}(\hat{\mathbf{Y}}_\theta, \tilde{\mathbf{Y}})$

Save teacher model as $f_{\theta^*}$

**Training Student GNN**:

Load teacher model $f_{\theta^*}$

**while** *Not converged* **do**

 | Teacher model conducts feature propagation on $\tilde{\mathcal{G}}$
 | **Optimization for** $\phi$
 | Student model conducts feature propagation on $\mathcal{G}$
 | Calculate reconstruction loss $\mathcal{L}_{rec}$ as in Eqn. (22)
 | $\phi \leftarrow \phi - \gamma_2 \nabla_\phi \mathcal{L}_{rec}$
 | **Optimization for** $\theta$
 | Calculate $\mathcal{L}_{pre}$ and $\mathcal{L}_{dis}$ as in Eqn. (23)
 | $\theta \leftarrow \theta - \gamma_1 \nabla_\theta \left( \mathcal{L}_{pre}(\hat{\mathbf{Y}}_{\theta^*}, \mathbf{Y}) + \alpha \mathcal{L}_{dis} \right)$

---

# G More Related Works

**Knowledge Distillation.** There are mainly four different types of distilling strategies [17], namely response-based KD [22] (which uses output layer of the teacher model to teach student), feature-based KD [42; 66; 21; 26] (which matches intermediate layers of teacher and student), relation-based KD [64; 65; 40; 62] (which aligns the relationship between different layers or samples), and graph-based KD [60; 63] (which considers the graph information or designed for GNN). For experiments, we choose representative method from each category as baselines. While these existing distillation strategies have shown remarkable success in distillation tasks such as model compression, they rarely (carefully) study the role of graph geometry in GNN iterations. For geometric knowledge transfer task, it is crucial to find a fundamental and principled way to track how graph topology affects the behavior of a specific GNN. Therefore, we first probe the intersection between KD and geometric learning, propose NHK to characterize geometric knowledge and propose different variants of GKD that are shown to be effective especially in the geometric knowledge transfer setting.

**Generalization of GNNs.** Recent advances shed lights on the generalization ability of GNNs from various perspectives, e.g., the in-distribution generalization error [16], extrapolation capability [59], out-of-distribution (OOD) generalization under distribution shifts [55] and the reliability against outliers and OOD testing data [32]. Our work can be seen a specific embodiment for GNN generalization w.r.t. topological domain transfer. Furthermore, there are some recent studies exploring learning under the varied data space between training and inference [54], using GNNs as an encoding and reasoning tool. GKD focuses on transferring across varied structural information and aims at compressing topological information for GNNs.