# OpenReview forum: "Geometric Knowledge Distillation: Topology Compression for Graph Neural Networks"
_NeurIPS.cc/2022/Conference — NeurIPS 2022 Accept_

### Official Review · Reviewer_1r32 · 2022-07-10

**Rating:** 7
**Confidence:** 4
**Soundness:** 3 good
**Presentation:** 3 good
**Contribution:** 3 good

**Summary:**

In this paper, the authors formalize the problem of graph topological knowlege transfer for GNNs. A new notion of neural heat kernel is proposed to characterize the geometric property of th eunderlying manifold for graphs. And a framework of geometric knowlege distillation to transfer geometric knowledge from a teacher GNN to student GNN.

**Questions:**

1. What's the effect of the proposed methoc on the social network dataset, e.g., Reddit?
2. There are many distilling strategies. Is it possible to give more comparison between different distilling strategies? And different parameters?


**Ethics Review Area:**

["I don’t know"]

**Limitations:**

The paper produces good experimental results and careful theoretical deriviation. I hope to see the effect on the dynamic graph of online distilling tasks.

**Strengths And Weaknesses:**

Strengths:
1. The paper proposes to distilling structural knowledge from teacher model of large graph to student model of small graph, which sounds novel and useful.
2. NHK is presented to encapsulate the geometric property of the underlying manifold concerning the architecture of GNN.
3. The experimental results demonstrate the efficacy of the proposed method.

---

> ### Author Response · Authors · 2022-08-02
> **Response to Reviewer 1r32**
>
> >***Q1: "What's the effect of the proposed method on the social network dataset, e.g., Reddit?"***
>
> New results on Reddit is shown in the following table, where our method still performs better than Teacher and close to Oracle. However, the reported performance may not be optimal as we didn’t have enough time to carefully fine-tune the method.
>
> |  | Oracle | Teacher | Student | KD | GKD-S | GKD-R |
> | --- | --- | --- | --- | --- | --- | --- |
> | Edge-aware | 89.3 | 81.8 | 73.6 | 80.5 | 85.7 | 86.8 |
> | Node-aware | 89.3 | 80.6 | 70.2 | 79.4 | 87.4 | 87.9 |
>
>
> >***Q2: "Is it possible to give more comparison between different distilling strategies? And different parameters?"***
>
> There are mainly four different types of distilling strategies [1], namely response-based KD [2] (which uses output layer of the teacher model to teach student), feature-based KD [3,4] (which matches intermediate layers of teacher and student), relation-based KD [5,6] (which aligns the relationship between different layers or samples), and graph-based KD [7,8] (which considers the graph information or designed for GNN). For experiments, we choose representative method from each category as baselines. While these existing distillation strategies have shown remarkable success in distillation tasks such as model compression, they rarely (carefully) study the role of graph geometry in GNN iterations. For geometric knowledge transfer task, it is crucial to find a fundamental and principled way to track how graph topology affects the behavior of a specific GNN. Therefore, we first probe the intersection between KD and geometric learning, propose NHK to characterize geometric knowledge and propose different variants of GKD that are shown to be effective, especially in the geometric knowledge transfer setting.
>
> In term of "parameters", we guess the reviewer may refer to the different parameter settings of teacher and student model. We have provided results for model compression in section 5.5 where student has different structures or sizes. **Similar results are also observed for other different parameter settings.**
>
>
> >***Q3: "The effect on the dynamic graph of online distilling tasks."***
>
> Online distillation in dynamic graphs is indeed an interesting application scenario where our method might be used. Unfortunately, we are not able to provide new results for this setting at this point since it is a new field for us and may require further modifications. Still, we provide new results for online distillation (where teacher and student are simultaneously trained) on Cora. Our model's performance is similar to that in the offline setting except for taking more time to converge (which is an intrinsic limitation of online distillation).
>
>  | Edge-aware  |Oracle | Teacher | Student | GKD-G | GKD-S | GKD-R | VGKD |
>  | -------- | -------- | -------- |-------- |-------- |-------- |-------- |-------- |
> |Offline  | $88.63\pm 0.48$ | $84.61\pm 0.37$ | $83.84\pm 1.32$  | $87.68\pm 1.07$     | $88.01\pm 0.79$     | $88.48\pm 0.59$ | $88.41\pm 0.62$  |
> |Online   | $88.63\pm 0.48$ | $84.61\pm 0.37$ | $83.84\pm 1.32$   |$87.75\pm 0.65$  | $87.63\pm 0.65$ | $88.28\pm 0.80$ | $88.50\pm 0.33$ |
>
>
>  | Node-aware |Oracle | Teacher | Student | GKD-G | GKD-S | GKD-R | VGKD |
> | -------- | -------- | -------- |-------- |-------- |-------- |-------- |-------- |
> |Offline  | $88.63\pm 0.48$ | $87.27\pm 0.51$ | $84.84\pm 1.61$  | $88.66\pm 0.85$  | $88.54\pm 0.52$     | $88.98\pm 0.39$ | $89.15\pm 0.45$  |
> |Online   | $88.63\pm 0.48$ | $87.27\pm 0.51$ | $84.84\pm 1.61$  | $88.32\pm 0.47$  | $88.40\pm 0.80$ | $89.00\pm 0.44$ | $88.92\pm 0.53$ |
>
> Anyway, we thank the reviewer for pointing this setting out, and we will consider it as a potential future research direction.
>
>
> [1] Knowledge Distillation: A Survey;
> [2] Distilling the knowledge in a neural network;
> [3] Fitnets: Hints for thin deep nets, in ICLR'15;
> [4] A comprehensive overhaul of feature distillation, in ICCV'19;
> [5] A gift from knowledge distillation: Fast optimization, network minimization and transfer learning, in CVPR'17;
> [6] Learning from multiple teacher networks, in KDD'17;
> [7] Tinygnn: Learning efficient graph neural networks, in KDD'20;
> [8] Distilling knowledge from graph convolutional networks, in CVPR'20;

---

> ### Author Response · Authors · 2022-08-06
> **A kind reminder before the discussion phase ends**
>
> Dear reviewer 1r32,
>
> Thanks again for your review. We hope our answers could increase your confidence. Since the discussion period is approaching its end, we would be glad to hear from you if our response has properly addressed your questions/concerns.
>
> Kind regards,
> The Authors

---

### Official Review · Reviewer_emui · 2022-07-11

**Rating:** 4
**Confidence:** 3
**Soundness:** 3 good
**Presentation:** 2 fair
**Contribution:** 2 fair

**Summary:**

The authors formulate a new knowledge distillation problem: making a GNN trained on a subgraph emulate the output of another GNN trained on a larger graph. To tackle the problem, they draw inspiration from thermodynamics to propose a kernel over GNN embeddings. Then, distilling knowledge is reduced to a task of matching kernel matrices --- or between a network's embeddings an another's smoothed by the kernel matrix.

**Questions:**

* Both parametric and non-parametric approaches depend on kernels, which might be a computational bottleneck. How do they scale with the number of nodes?

**Limitations:**

Paper lacks a proper discussion of its own limitations. There is also no statement regarding possible societal impacts.

**Strengths And Weaknesses:**

## Strengths
* To the best of my knowledge, authors draw a novel connection between graph convolutions and heat kernels;
* Authors attack an original problem: condensing the geometry of a large graph into a GNN trained on a subgraph;

## Weaknesses
* While novel, the motivation for the problem is weak. In most interesting cases, we will only observe a subgraph and there is no oracle capable of outputting embeddings based on the whole graph. This seems to limit the applicability of this sort of method. it would be more reasonable just to have some other graph(s) independently drawn from the same data-generating process;
* The connection to thermodynamics seems very thin, as it just reduces to computing kernels over embeddings.

---

> ### Author Response · Authors · 2022-08-02
> **Response to Reviewer emui**
>
> > ***Q1: "While novel, the motivation for the problem is weak. In most interesting cases, we will only observe a subgraph and there is no oracle capable of outputting embeddings based on the whole graph. This seems to limit the applicability of this sort of method."***
>
> In the following, we summarize our motivation from both theoretical and practical points of view, and also show that the applicability of our method is beyond the geometric knowledge transfer setting.
>
> **1. Theoretical motivation and contributions**
>
> **(Motivation and contribution)** As mentioned in the introduction, as an exploratory work, this paper is motivated for answering the question of "whether it is possible to use graph topology as a special type of geometric prior" in the background of geometric deep learning [1]. We think this step is important for the development of the field and will provide insights for developing new GNN paradigms where graph topology is not treated as input.
>
>
> **(More contribution)** As a general theoretical tool, the formulated NHK provides a new perspective to the role of graph topology in characterizing underlying geometry, which itself is valuable for understanding GNNs and may inspire other theoretical works in graph learning field.
>
>
> **2. Application for geometric knowledge transfer (GKT)**
>
> The geometric knowledge transfer problem we consider in this paper incorporates a class of well-established problems studied in different fields, which however lacks unification and in-depth theoretical investigation. Our work fills this particular research gap. To list a few application scenarios:
>
> * Inference efficiency [2]: tries to coarsen the graph structure to boost inference speed to deploy for latency-constraint applications that require fast inference.
> * Recommender systems [3]: tries to condense knowledge from heterogeneous graph structure (with different types of edges) to a sub-graph with only one type of edge.
> * Federated learning [4]: considers a setting in federated learning where each local system holds a small subgraph that could not be directly shared.
>
>
> **3. Application for conventional knowledge distillation**
>
> Admittedly, application for GKT alone is relatively limited. This is why we also conduct experiments in Section 5.5 to show that our method is also effective for widely-recognized conventional KD purposes (i.e., model compression and self-training) with indisputably large audience. This fact combined with the application for GKT makes our method useful in broader areas.
>
>
> > ***Q2: "The connection to thermodynamics seems very thin, as it just reduces to computing kernels over embeddings."***
>
> We respectfully disagree with this statement as our implementations are either directly built upon (i.e., VGKD, GKD-R), or guided and explained by (i.e., other non-parametric GKD) our theories derived from thermodynamic and geometric perspectives.
>
>
>
> **(Non-parametric Case)** Theorem 1/2 explains non-parametric GKD is essentially matching the underlying manifolds of two GNN models, while the chosen kernel characterizes the geometry of these manifolds. For example, GKD with Gauss-Weierstrass kernel can be explained as matching latent manifolds in Euclidean space. Further, GKD-R is supported by Theorem 3 which is also derived from thermodynamics, and it has indeed shown to be effective.
>
> Non-parametric GKD possesses a simple form in order to be easily implemented and applied in real-world scenarios. Despite its simplicity, it is theoretically grounded so that we are able to answer some critical questions about which design is more appropriate and why it can work.
>
> **(Parametric Case)** Theorem 2 induces NHK across multiple layers of GNN, and directly leads to the formulation of VGKD. This advanced variant aims to explore the true NTK behind the GNN model and match their manifolds in the exact latent space. The detailed training procedure is described in Eq.(21-23) and also theoretically justified by Proposition 1. This implementation goes far beyond “just computing kernels over embeddings”.
>
>
> > ***Q3: "Both parametric and non-parametric approaches depend on kernels, which might be a computational bottleneck. How do they scale with the number of nodes?"***
>
> Please refer to the general response titled "How we scale to large graphs" at the top where we show our method can seamlessly integrate with mini-batch sampling methods used to train large graphs without compromising on performance.
>
> >***Q4: "Paper lacks a proper discussion of its own limitations."***
>
> For clarity, we list and summarize the limitations and potential societal impacts in the last section of Appendix in our new version.
>
> References
> [1] Geometric Deep Learning Grids, Groups, Graphs, Geodesics, and Gauges;
> [2] Faster graph embeddings via coarsening, ICML’20;
> [3] Privileged graph distillation for cold start recommendation, SIGIR’21;
> [4] Subgraph Federated Learning with Missing Neighbor Generation, NeurIPS’21

---

> ### Author Response · Authors · 2022-08-06
> **A kind reminder before the discussion phase ends**
>
> Dear reviewer emui,
>
> Thanks again for your review. We have provided informative answers to clarify our theoretical motivation, practical applicability, scalability, and close connection between implementation and theories. We hope these answers could be valuable for your re-assessment of our work.
>
> Since the discussion period is approaching its end, we would be glad to hear from you if our response has properly addressed your questions/concerns.
>
> Kind regards,
> The Authors

---

### Official Review · Reviewer_1tL3 · 2022-07-18

**Rating:** 5
**Confidence:** 4
**Soundness:** 3 good
**Presentation:** 2 fair
**Contribution:** 4 excellent

**Summary:**

This paper proposes a new method of knowledge distillation for graph neural
network, which takes into account the geometry of the network architecture, in
particular edge- and node-awareness. The objective of knowledge distillation in
the context of teacher-student graph networks is to try to make the student
networks, trained on partial graph (hence much smaller scale), mimic the best
classification performance of the teacher (trained on complete graph).

To do this, the authors propose a generalization of the heat kernel for GNNs,
dubbed Neural Heat Kernel (NHK), based on the previous works that established
connection between heat kernels and different GNNs architecture. They establish
theoretical arguments for the existence of a single-layer NHK (theorem 1), and
a semigroup identity property (theorem 2) which proves that the operator
resulting from stacking multiple later of NHKs remains a valid kernel.

In terms of modeling, they proposed the non-parametric and parametric form of
NHKs, together with corresponding loss functions and training schemes. The
non-parametric distillation compose of three different forms of deterministic
neural heat kernels, while parametric is based on an EM-like training
scheme. All of these are demonstrated to have better classification
performances from related methods and even oracle networks.


**Questions:**

- In theorem 1 regarding the existence of NHK, I see the assumtion to have the
  equivalence between graph signal X and node features layer H. Can the authors
  comment more on this assumption, for example, will this ever be violated?

- In Section 5.4 on performance with privileged ratio, maybe I'm missing
  something but why do we have the oracle in edge-aware setting perform quite
  bad in only Pubmed dataset (i.e. I don't get your point on the
  justification)?


**Limitations:**

Limitations have not been discussed in the main text.

**Strengths And Weaknesses:**

Strengths:

- Overall I find the paper well-written, with clear structure -- although a bit
  dense and undefined notations in the beginning (comments below in the
  weakness).

- The paper propose a more rigorous formalization of Knowledge Distillation in
  the context of Geometric Deep Learning, with theoretical justification borrow
  from PDE and differential geometry literature.

- The authors deliver some theoretical arguments to their proposed methods, and
  empirical results show they all perform better than existing solutions on
  even large-scale dataset.

Weaknesses:

- Major: I see absolutely no discussion on the computational time of the NHKs
  in both parametric and non-parametric settings. Since now the training loss
  need to take into account additional information of the underlying
  topological space, the training schemes look quite complicated, which require
  heavier part on hyperparameters-tuning. The choice of hyperparameters are
  addressed in the Appendix, which resort to grid-search, so if I'm not wrong
  this does not scale well in terms of computational time. I insist the authors
  provide justification on this part, and maybe even better to add it in the
  main text.

- Major: related to first point, even though the performance of parametric NHK
  (denoted by VGKD in the experimental results section) is generally the best,
  I see no discussion on the convergence speed EM-like scheme for training the
  parametric NHK.

- Minor: The matching loss $\mathcal{L}_{pre}$ is nowhere to be found in the
  paper, which render missing equation in algorithm box 2. Could the author
  provide the form of this loss somewhere, as it is quite important part as
  well?

- Minor: I'm a bit confused about the notation of the Example 1 in Section 2 --
  what is $\sigma$ fuction and what is $\mathbf{\Omega}$? Also is this the same
  $\sigma$ to denote activation function, which is used in Eq.(16)?

---

> ### Author Response · Authors · 2022-08-02
> **Response to Reviewer 1tL3**
>
>
> >***Q1: "Discussion on the computational time of the NHKs in both parametric and non-parametric settings"***
>
> Please refer to the general response titled "How we scale to large graphs" at the top where we analyzed the computation complexity, and show how our method can seamlessly integrate with mini-batch sampling methods used to train large graphs without compromising on performance. We also want to further remark that, in practice, training in both parametric and non-parametric settings could be done on a normal GPU within a reasonable amount of time (several minutes for small graph, and less than one hour for large graph on our device).
>
> >***Q2: "The training schemes look quite complicated, which require heavier part on hyperparameters-tuning"***
>
> Note that table.9 in appendix list hyperparameters for all variants. There are at most 4 hyperparameters (GKD-R) for a single variant. In practice, the most important hyper-parameter is the distillation loss weight $\alpha$, which has already been discussed in table.10. Our model is less sensitive to other hyper-parameters. Therefore, instead of grid search, one can use other more efficient searching strategies to achieve very similar results.
>
> > ***Q3: "Discussion on the convergence speed EM-like scheme for training the parametric NHK"***
>
> We compare the average epoch (in five runs) for reaching convergence of different models under the same hyper-parameter setting. The results are:
>
> |  | Teacher / Oracle | Student | GKD(-S) | VGKD |
> | --- | --- | --- | --- |  --- |
> | Cora | 21.6 | 13.8 | 46.2 | 71.4 |
> | Citeseer | 17.0 | 11.4 | 33.4 |  52.6 |
> | Pubmed | 90.2 | 65.4 | 61.6 | 102.0 |
>
> We can see that in general VGKD takes more (but not much more) epochs to converge compared with the non-parametric variant. The result is explainable by noting that the learnable mapping $g_\phi$ used in Eq.(22) is an independent module since it only "calibrates" the distillation loss but does not affect GNN's feed-forward computation.
>
> We also want to remark that the model needs not to wait for Eq.(22) to converge to achieve good performance (though this step is necessary for further improvement) considering the fact that even using a random mapping function (i.e., GKD-R) is effective.
>
> > ***Q4: "The matching loss $\mathcal{L}_{p r e}$ is nowhere to be found in the paper, which render missing equation in algorithm box $2.$"***
>
> Fixed in the revised version. $\mathcal{L}_{p r e}$ refers to the first term in Eq.17, which denotes prediction loss used for classification or regression.
>
> > ***Q5: "notation of the Example 1 in Section 2: what is $\sigma$ function and what is $\boldsymbol{\Omega}$ ? Also is this the same $\sigma$ to denote activation function, which is used in Eq.(16)?"***
>
> The $\sigma$ function in example 1 denotes softmax function, and $\sigma$ in Eq.(16) denotes general activation function. To keep notation consistent, we changed $\sigma$ in example 1 to $Softmax$. And $\boldsymbol{\Theta} \in \mathbb R^{d\times k}$ in example 1 denotes a learnable transformation matrix as model parameter. Thanks for pointing them out.
>
>
> > ***Q6: "In theorem 1 regarding the existence of NHK, I see the assumption to have the equivalence between graph signal X and node features layer H. Can the authors comment more on this assumption, for example, will this ever be violated?"***
>
> The short answer is that the assumption strictly holds for some specific GNN architectures and is unknown (i.e. hard to prove or disprove) for other GNNs, but using it for theoretical analysis is reasonable enough based on existing evidence suggesting their equivalence. Please refer to the response to Q3 reviewer w36o (the first reviewer), where we discussed "why the two expressions in Eq.7 could be equivalent", for more detailed explanations.
>
>
> > ***Q7: "Why do we have the oracle in edge-aware setting perform quite badly in only Pubmed dataset?"***
>
> We guess the reviewer intended to refer to the teacher model, which indeed performs badly on Pubmed, instead of Oracle (in Fig.2.c). This is because the teacher model is trained on the complete graph (which is presumed to be unavailable for test), and tested in the partial graph. We conjecture that the privileged part of graph information is crucial for the teacher model to perform well in Pubmed (compared with other datasets), which explains its decreasing performance w.r.t. larger PIR.

---

> > ### Comment · Reviewer_1tL3 · 2022-08-06
> > **Thank you for your detailed rebuttal and revision, but I found it unsatisfactory**
> >
> >
> > I thank the authors for their time and efforts to addressed reviewers' concerns (and my apologies for the delayed response), with an addition of extra experiments.
> >
> > At the current state of revision, __I do not think my major concerns (about the scalability of the method and a more thorough discussion of convergence of the method in the main text) are addressed__. The lack of discussion on limitation of the methods inside the main text makes it felt incomplete and lack the transparency, and I do not find the addition of 6 lines on the limitation, mainly about computational time/scalability at the very end of the Appendix satisfactory. Moreover, I suggest the authors adding the table on average epochs to reach convergence inside the Appendix, together with the general responses on how did they scale large graph.
> >
> > >We compare the average epoch (in five runs) for reaching convergence of different models under the same hyper-parameter setting. The results are:
> >
> >  	    Teacher / Oracle 	Student 	GKD(-S) 	VGKD
> >         Cora 	21.6 	13.8 	46.2 	71.4
> >        Citeseer 	17.0 	11.4 	33.4 	52.6
> >        Pubmed 	90.2 	65.4 	61.6 	102.0
> >
> > I'm not sure I get the point, but regarding the scalability, it is clear from the additional table you shown in your reponse that both the parametric and non-parametric computational time poorly. A less major concern, mainly due to that my understanding about the lack of time, is that this is computational time on average of only 5 runs (without standard deviation). I also don't get the point why the authors did not report avg. wall-clock-time per  epochs as an addition to this table.
> >
> >
> > Finally, in the Limitations (section G in Appendix), the authors wrote:
> >
> > > One of the main limitations in theoretical analysis is using the equivalence that connects GNN and heat diffusion on manifold, __as it holds only for some specific GNN instantiations__. However, __this perspective is fundamental and useful for studying the geometry of GNNs, which has also been adopted by other works__ to produce meaningful theoretical results and insights.
> >
> > This lacks elaborations on two points, which have been highlighted:
> > - Which kind of "specific instantiations" the authors want to mention in the sentence?
> > - The second highlighted part need citations (akin to what the authors have replied to one of the reviewers). Can the authors add the reponses to Q3 of Reviewer w36o as a Remark somewhere near Theorem 1?

---

> > > ### Author Response · Authors · 2022-08-07
> > > **Further response to Reviewer 1tL3 (Part 1 of 2)**
> > >
> > > We greatly thank you for the valuable feedback that can help us for further improvement. We have significantly modified our paper (see blue parts in the new version) and add more discussions below with new results to resolve your concerns.
> > >
> > > ### More scalability discussions
> > > > **Q1: "I do not find ... at the very end of the Appendix satisfactory.", "I suggest the authors adding the table inside the Appendix, together with how did they to scale large graph."**
> > >
> > > We have added Appendix H to include all the results of scalability and explanation of "how our method scales to large graph" in the newly uploaded paper. Due to the limited time, we will try our best to refine this part in the final version with more extensive empirical results, detailed descriptions and plots.
> > >
> > > > **Q2: "it is clear from the additional table you shown in your reponse that both the parametric and non-parametric computational time poorly."**
> > >
> > > The central message we are to convey from the original table is that **1) both proposed models can converge within reasonable number of epochs (less than 200) and 2) the EM-style algorithm does not cost too much extra epochs for convergence than the non-parametric model**. Such a result is important since we usually use a fixed epoch number (i.e., 200 for citation networks) as the default setting for training as recommended in [1].
> > >
> > > We'd like to point out that in broader context, existing KD methods [2-5] naturally cost more time than the non-KD counterpart (the teacher and student models) due to additional distillation loss. This is also amplified in our geometric knowledge transfer setting because of the inconsistency of graph topology that makes it harder for student to mimic the behavior of teacher. Therefore, in our work, it is normal that the proposed GKD costs more time than the pure teacher/student models. Such extra time is often acceptable and unimportant in this area due to that the more important factors of interest are usually the inference efficiency and effectiveness of student model [2-5].
> > >
> > > As further evidence to resolve the potential concern towards the efficiency/scalability, we supplemented more comparisons with other peer KD methods in terms of convergence epochs (we add standard deviation in original result, and use more runs in new results).
> > >
> > > |  | Teacher / Oracle | Student | GKD(-S) | VGKD | KD| FitNets |FSP |LSP|
> > > | --- | --- |  --- | --- |  --- | ---|---|---|---|
> > > | Cora | $21.6\pm 6.1$| $13.8\pm 3.7$| $46.2\pm 7.2$ | $71.4\pm 10.1$ |$38.4\pm 5.4$ |$62.5\pm 10.7$ | $94.2\pm 12.6$ |$41.5\pm 7.4$ |
> > > | Citeseer | $17.0\pm 4.6$ |$11.4\pm 6.0$ | $33.4\pm 9.2$ |  $52.6\pm 11.2$ |$29.0\pm 8.3$ |$48.8\pm 12.9$ |  $72.6\pm 12.0$ |$31.3\pm 5.4$ |
> > > | Pubmed |$90.2\pm 17.2$|$65.4\pm 14.8$| $61.6\pm 15.6$ | $102.0\pm 12.5$ |$60.6\pm 12.8$ |$94.4\pm 20.6$ | $122.4\pm 24.5$ |$70.1\pm 14.0$ |
> > >
> > > We can see that the convergence speed of our models are on par with other KD baselines, while outperforming them significantly in terms of testing performance in geometric knowledge transfer settings (as shown in Table 1,2,3 in our paper).
> > >
> > >
> > > > **Q3: "the authors did not report avg. wall-clock-time per epochs as an addition to this table"**
> > >
> > > Thank you for the nice suggestion that helps to strengthen our discussions. Since again in practice we use a fixed number of training epochs (e.g., 200 for citation networks) for all the models, the training time is proportional to the time per epoch, which is reported in the following table:
> > >
> > > |  | Teacher / Oracle | Student | GKD(-S) | VGKD | KD| FitNets |FSP |LSP|
> > > | --- | --- | --- | --- |  --- |--- | --- | --- | --- |
> > > | Cora (ms) | 37.2 | 35.8 | 58.7 | 73.4 | 41.3 | 55.2 | 78.7| 60.0 |
> > > | Citeseer (ms) | 36.6 | 32.9 | 49.8 |  66.7 | 42.5 |47.3 | 70.2| 57.2 |
> > > | Pubmed (ms) | 53.9 | 43.6 | 75.3 | 92.4|61.5| 68.1|101.0|80.6|
> > > | OGB-Arxiv (s) | 2.43 | 2.30 | 2.91 | 3.38 | 2.62 | 2.75 | 3.64 |3.06
> > > | OGB-Proteins (s) | 4.46 | 4.25 | 6.05 | 7.90 | 4.89 | 5.35 | 8.41 | 6.22
> > >
> > > Remarkably, our parameteric version in the worst case takes less than two times of the training time of vanilla KD and is even more efficient than the relation-based KD method FSP. In practice, the training in citation networks (in one run) can be done within a minute, which is very efficient. Also, our models successfully scale to large OGB datasets (more than 100,000 nodes) with comparable time as other models and yielding significantly better performance, which again verifies the efficacy of our models.

---

> > > > ### Author Response · Authors · 2022-08-07
> > > > **Further response to Reviewer 1tL3 (Part 2 of 2)**
> > > >
> > > > > **More Justifications**
> > > >
> > > > We also want to remark that the training time cost is not a major factor of interest in research for KD, and it is seldomly discussed in existing KD papers [2-5]. There are also more complicated approaches that are still widely adopted in practice, such as using multiple teacher models (e.g., [6]), adversarial learning (e.g., [7]), and multi-stage distillation (e.g., [8]). By contrast, our model remains simple and efficient, yet it shows promising power for GKD purposes.
> > > >
> > > > Instead of training time cost, the key question in our context is really about whether we can scale the training to larger graphs, which is addressed by our general response "how our model scales to largers graphs" and verified by the experiment. In comparison, to our knowledge, existing graph-based KD methods [9-12] have almost never studied graphs with more than 100,000 nodes as we did in experiments.
> > > >
> > > >
> > > >
> > > > [1] Semi-supervised classification with graph convolutional networks, in ICLR'17;
> > > > [2] Knowledge distillation: A survey;
> > > > [3] Distilling the knowledge in a neural network;
> > > > [4] Fitnets: Hints for thin deep nets, in ICLR'15;
> > > > [5] A Gift from Knowledge Distillation: Fast Optimization, Network Minimization and Transfer Learning, in CVPR'17;
> > > > [6] Learning from Multiple Teacher Networks, in KDD'17;
> > > > [7] Kdgan: Knowledge distillation with generative adversarial networks, NeurIPS'18;
> > > > [8] Improved knowledge distillation via teacher assistant, AAAI'20;
> > > > [9] Extract the Knowledge of Graph Neural Networks and Go Beyond it: An Effective Knowledge Distillation Framework, in WWW'21;
> > > > [10] TinyGNN: Learning Efficient Graph Neural Networks, in KDD'20;
> > > > [11] Distilling Knowledge from Graph Convolutional Networks, in CVPR'20;
> > > > [12] Graph Few-Shot Learning via Knowledge Transfer, in AAAI'20;
> > > >
> > > > ## Modification for the paper on limitation section and Thm 1
> > > >
> > > > Thank you for the nice suggestions. We have modified our paper with the corresponding parts colored blue.
> > > >
> > > > - In the para 2 of Appendix I, which is now the limitation section in the new version, we add more details for explaining the equivalence in Eq.7, including specific GNN instantiations (e.g., SGC, APPNP, GAT) and references that adopt similar perspective for theoretical analysis. We also supplement more concrete contents in para 2/3 in Appendix G and outlook to strengthen this part for better readability and insightful presentation.
> > > > - In the para 3 of Appendix I, we discuss the potential limitation concerning time costs and scalability, and also summarize our results and effective ways for scaling to large graphs, as detailedly illustrated in Appendix H.
> > > > - We also modified Thm 1 with a new pointer to Appendix G where we present more detailed elaboration on the equivalence property. We will move more content to the main text (e.g., remark below Thm 1, descriptions in Appendix H) once we have one extra page in the camera-ready version.
> > > >
> > > > We hope our new response is helpful for addressing your potential concerns. If you have any further question/feedback, please let us know.

---

> ### Author Response · Authors · 2022-08-06
> **A kind reminder before the discussion phase ends**
>
> Dear reviewer 1tL3,
>
> Thanks again for your review. We hope our answers and explanations could increase your confidence. Since the discussion period is approaching its end, we would be glad to hear from you if our response has properly addressed your questions/concerns.
>
> Kind regards,
> The Authors

---

### Official Review · Reviewer_w36o · 2022-07-20

**Rating:** 6
**Confidence:** 3
**Soundness:** 3 good
**Presentation:** 3 good
**Contribution:** 2 fair

**Summary:**

This paper focus on the knowledge distillation of GNN, where the teacher GNN has access to the complete version of a (sometimes large) graph, and the student GNN is only exposed to parts of it. The authors introduce the concept of heat kernels into the GNN formulation (termed "neural heat kernels"), given that it has been utilized in graph applications before GNN and has demonstrated its experimental effectiveness in the context of graph distillation.

**Questions:**

I am unfamiliar with this research topic; therefore, I would like to see the discussions on the following points.

**[Why the two expressions in Eq.7 could be equivalent, i.e., Spatial vs Time]**

**[Depth vs Width, in the formulation of NHK]**

**[Computation Complexity]**


**Limitations:**

The authors claimed to address the limitations of their work in the checklist, however, I can't find its location. Adding the pointer to describe the limitations and future directions would be better.

**Strengths And Weaknesses:**


- ***Originality***: The methods are newly formulated, the existed task of grpah distiallation is well defined.The related work is adequately cited.

- ***Quality***: The work is complete, my concerns on the experimental settings and results are put at the *clarity* and **Questions**.

- ***Clarity***: The submission is clearly written and well organized. My concern is why NHK is especially helpful for the setting of graph distillation compared with other methods that describe the latent structure of the graphs. Also:

  **[unify the notations]** e.g., in line 124 and eq.7, $\Delta(\theta, G)$ and $\Delta\left(f_{\theta}, \mathcal{G}\right)$ are essetially representing the same entity, a Laplace-Beltrami operator.

- ***Significance:*** With the code provided in the supplementary, other researchers or practitioners can use or build on the ideas. However, the performance gains are insignificant since the benchmark (Cora, Citeseer, and PubMed) are well studied and have less space for improvement.

---

> ### Author Response · Authors · 2022-08-02
> **Response to Reviewer w36o (Part 1 of 2)**
>
> > ***Q1: "Why NHK is especially helpful for the setting of graph distillation compared with other methods that describe the latent structure of the graphs."***
>
>
> For short, this is because NHK is more like "a reflection of how a GNN utilizes the graph throughout the entire message passing process" than a “method that describes the latent structure of the graph”. The latter is related to graph structure learning and is insufficient to describe the behavior of GNNs. Differently, NHK is derived from a more principled aspect of the geometric property of graphs and the knowledge GNNs extract. We next briefly explain why we consider NHK in geometric knowledge transfer.
>
> The foremost challenge of successful geometric knowledge transfer (so-called “graph distillation”) is finding a principled and fundamental way to encapsulate the knowledge extracted by GNN model from the graph topology. However, it is hard to directly track how graph topology affects the behavior of a specific GNN. That’s why we view GNNs from a thermodynamic perspective where graph convolution is equivalent to heat diffusion on the underlying manifold (see more discussions in Q3). With this connection, we reduce the question to characterizing the geometric property of this manifold (which determines the diffusion process, namely the behavior of GNN). This leads to the formulation of NHK based on reasonings given in section 2 and 3.1 about heat kernel.
>
> By means of NHK, the distillation task can be resolved by transferring the geometric knowledge between GNN models on different graphs. Compared to other works on latent structure that purely leverage edge-level information, NHK enables knowledge transfer through a more in-depth common nature, i.e., the global geometric property, yielding better representational expressiveness and model capacity.
>
> Moreover, beyond distillation tasks, this result (NHK as characterization of GNN) alone is of independent interest in geometric deep learning, and may hopefully inspire other theoretical works for understanding GNNs.
>
> > ***Q2: "$\Delta(\theta, G)$ and $\Delta\left(f_{\theta}, \mathcal{G}\right)$ are essetially representing the same entity, a Laplace-Beltrami operator"***
>
> We have unified them as $\Delta\left(f_{\theta}, \mathcal{G}\right)$. Thanks for pointing it out.
>
> > ***Q3: "Why the two expressions in Eq.7 could be equivalent, i.e., Spatial vs Time"***
>
> The equivalence of two equations in Eq.7 is based on recent works [1-5] that built a connection between the heat equation and GNN. The main result of these works is: *"By treating node features H as signal X (corresponding to $x(u,t)$ in heat equation Eq.2) on the graph, solving the heat equation with Euler scheme yields the formulation of a GNN layer."*
>
> In other words, **the GNN can be seen as the discretizations of the continuous diffusion process described by the heat equation**. Correspondingly in Eq.7, the left equation is a general GNN layer (discretized diffusion process), and the right equation is directly derived from Eq.2 (continuous diffusion process).
>
> Moreover, different definitions of Laplace-Beltrami operator $\Delta$ yield different GNNs (such as SGC and linear GAT). Unfortunately, not all GNNs have a simple form of $\Delta$, and for some of them, whether there exists such $\Delta$ is an open research question. Therefore, we write the operator as $\Delta(f_\theta, \mathcal G)$ to associate it with model $f_\theta$ and use equivalence in Eq.7 as an analytical assumption.
>
>
> References:
> [1] GRAND: Graph Neural Diffusion, in ICML’21;
>
> [2] Beltrami Flow and Neural Diffusion on Graphs, in NeurIPS’21;
>
> [3] Dissecting the Diffusion Process in Linear Graph Convolutional Networks, in NeurIPS’21;
>
> [4] PDE-GCN: Novel Architectures for Graph Neural Networks Motivated by Partial Differential Equations, in NeurIPS’21;
>
> [5] Graph Neural Networks as Gradient Flows, arxiv’22

---

> > ### Author Response · Authors · 2022-08-02
> > **Response to Reviewer w36o (Part 2 of 2)**
> >
> > > ***Q4: "Depth vs Width, in the formulation of NHK"***
> >
> > Since the meaning of comparing "depth" and "width" is unclear, we will next discuss how they impact the formulation of NHK (theoretically and practically) in the following.
> >
> > **(Depth)** As described in section 3.1, the definition of NHK is associated with two layers ($l$ and $l+k$) in the GNN. For implementation, we choose $k=1$ for non-parametric NHK based on the fact that multi-layer distillation loss is equivalent to the single-layer distillation loss by changing hyper-parameters. For parametric NHK, we choose $l+k$ as the last layer and $l$ as the first layer in order to learn a NHK that uses as much information as possible.
> >
> > **(Width)** We guess the reviewer may refer to the graph size or the dimension of node features as we don’t recall mentioning the width of NHK. If this is the case, then both factors could affect the formulation of NHK by changing the definition of $\Delta(f_\theta, \mathcal G)$. Empirically, our method is equally effective across different settings of both factors.
> >
> > > ***Q5: "Computation complexity"***
> >
> > The short answer is $O(nd^2)$. Please refer to the general response titled "How we scale to large graphs" at the top where we analyzed the computation complexity and show our method can seamlessly integrate with mini-batch sampling methods used to train large graphs without compromising on performance.
> >
> >
> > > ***Q6: "Adding the pointer to describe the limitations and future directions would be better."***
> >
> > For clarify, we list and summarize the limitations and potential societal impacts in the last section of Appendix in our new version.
> >
> >
> > > ***Q7: "The performance gains are insignificant since the benchmark (Cora, Citeseer, and PubMed) are well studied and have less space for improvement."***
> >
> > We agree with the reviewer in that gaining additional performance on the benchmark is hard. Nevertheless, as an exploratory work on the possibility of injecting geometric knowledge, this paper focuses on relative performance comparison rather than absolute performance gain. Notably, our method consistently exceeds teacher model and even rivals with the oracle model despite using less graph information that has been conventionally considered as crucial for GNN's good performance. The reason why we choose the citation networks as the experimental datasets is that they are the most commonly used benchmarks for developing graph-based models and are easy for calibration with existing works.

---

> ### Author Response · Authors · 2022-08-06
> **A kind reminder before the discussion phase ends**
>
> Dear reviewer w36o,
>
> Thanks again for your review. We hope our answers and explanations could help you better evaluate our work. Since the discussion period is approaching its end, we would be glad to hear from you if our response has properly addressed your questions/concerns.
>
> Kind regards,
> The Authors

---

> > ### Comment · Reviewer_w36o · 2022-08-08
> > **Thanks and sorry for the late.**
> >
> > Thanks for your response. Most of my concerns are addressed. I raised my score as leaning more for acceptance :)

---

### Author Response · Authors · 2022-08-02
**General Response: How we scale to large graphs**

Suppose the size of the partial graph (i.e., the small one) is $n$. The computation complexity for distillation loss (in both non-parametric and parametric cases) is $O(dn^2)$, and the extra space consumption is $O(n^2)$.

Directly using the whole graph is impractical when the partial graph is large. Therefore, as mentioned in appendix E.2, we recommend using mini-batch training for large graphs, which has already been adopted for training two large-scale datasets in our experiment. **It could also be seamlessly integrated with the original mini-batch method (by sampling ego-graphs) used for large graphs without further modification.** Our method could be run on most GPUs within a reasonable amount of time. Moreover, we want to remark on the following points:

1. The distillation loss for mini-batch training is **NOT biased** as long as nodes are evenly sampled, and thus **will not harm the performance**. We further support this by supplementary experiments (in Cora and edge-aware setting) that show GKD's performance with decreasing batch size:

| Batch Size | Whole graph | 1024 | 512 | 256 | 128 |
| --- | --- | --- | --- | --- | --- |
| GKD(-R) | $88.48\pm 0.59$ | $89.02\pm 0.52$ | $88.48\pm 0.30$ | $88.63\pm 1.07$ | $88.28\pm 0.30$ |
| VGKD | $88.41\pm 0.62$ | $88.72\pm 0.53$ | $88.82\pm 0.37$ | $88.47\pm 0.31$ | $88.18\pm 0.96$ |

2. Notably, the distilled student model is **efficient for inference** because of the sparser graph structure and perhaps smaller model size, which is probably a more important factor than training efficiency for those who resort to KD.
3. Even in the case where we somehow must use the whole large graph for calculating the loss (which is extremely rare or doesn't exist), there are **ways to reduce the complexity** (e.g., pre-computing teacher's NHK matrix, using low-rank approximation).

---

### Author Response · Authors · 2022-08-02
**Message to area chairs and all the reviewers**

We appreciate the reviewers' time, valuable feedback and constructive suggestions. We are delighted that reviewers deem our paper well-written and our methods novel (w36o, emui, 1r32), rigorous (1tL3), well defined (w36o), and useful (1r32). The reviewers also reckoned "other researchers or practitioners can use or build on the ideas" (w36o) and appreciated our theoretical results that build a "connection between graph convolutions and heat kernels" (emui). We address all the questions with newly supplemented experiment results in each response and welcome further comments or requests.

---

### Author Response · Authors · 2022-08-07
**Modification and New Draft Uploaded**

Dear Reviewers,

We again appreciate your time and valuable feedback. Based on the initial review comments, we have significantly improved the draft with a new version uploaded. For your reference, we mark the modified parts with blue color and summarize the key points below:

- For **Reviewer w36o**: Q1 in line 33-35, 46-49; Q2 in Eq.7, Q3 in Appendix G (pointer in thm 1); Q4 in line 228-229; Q5 in Appendix H.1 (pointer in line 245-246);  Q6 in appendix I (pointer in conclusion).
- For **Reviewer 1tL3**: Q1 in Appendix H.3; Q2 in Appendix H.2 (pointer in line 245-246); Q3 in Appendix F.2; Q4 in Alg. 1 and 2; Q5 in example 1; Q6 in Appendix G (pointer in thm 1); Q7 in Section 5.4.
- For **Reviewer emui**: Q1 in line 49-51, 63-64; Q3 in Appendix H (pointer in line 245-246); Q4 in Appendix I (pointer in conclusion).
- For **Reviewer 1r32**: Q1 in Appendix F.3; Q3 in Appendix F.4 (pointer in line 245-246); Q2 in Appendix E.5 (pointer in Section 1.2).

We hope the new version could facilitate the reviewing process toward a more comprehensive evaluation of our work. If there needs any further changes, please let us know during the discussion phase.

---

### Meta-Review · Area_Chair_UykR · 2022-08-25

**Recommendation:** Accept
**Confidence:** Certain

**Metareview:**

In this paper the authors propose the first distillation method for GNN to promote  the transfer of knowledge from a large graph to a smaller graph. The authors propose an extension of Neural Heat Kernel to encode information in the GNN and use it to propose  Geometric Knowledge Distillation. Numerical experiments are done with parametric and non-parametric versions of the method and show its interest on several tasks such as node classification, and more classical distillation tasks such as model compression and self distillation.

The contribution was appreciated and all reviewers agree about the novelty of the connection between tangent kernel and GNN, and its possible use in other applications in the future. The experiments are interesting and often show a large gain in performance wrt classical distillation methods. Some concerns were raised by reviewers: numerical complexity, limited practical interest, lack of discussion about limits.

Most of the concerns have been answered quite well by the authors during the discussion which was much appreciated.  The precise tables with numerical complexity and the limitation discussion now in appendix are very interesting. So the consensus among reviewers was that the paper deserves publication but the numerical complexity and complexity discussion MUST be integrated in the main paper instead of being in supplementary.

**Award:**

No

---

### Decision · Program_Chairs · 2022-09-14

Accept